# A translational regulator MHZ9 modulates ethylene signaling in rice

Yi-Hua Huang [1,6], Jia-Qi Han [1,2,6], Biao Ma[3,6], Wu-Qiang Cao[1,2], Xin-Kai Li[1,2], Qing Xiong [4], He Zhao[1], Rui Zhao[1,2], Xun Zhang[1,2], Yang Zhou [1], Wei Wei[1], Jian-Jun Tao[1], Wan-Ke Zhang [1], Wenfeng Qian [1,2], Shou-Yi Chen [1], Chao Yang [5] ✉, Cui-Cui Yin [1] ✉ & Jin-Song Zhang [1,2] ✉

Ethylene plays essential roles in rice growth, development and stress adaptation. Translational control of ethylene signaling remains unclear in rice. Here, through analysis of an ethylene-response mutant *mhz9*, we identified a glycine-tyrosine-phenylalanine (GYF) domain protein MHZ9, which positively regulates ethylene signaling at translational level in rice. MHZ9 is localized in RNA processing bodies. The C-terminal domain of MHZ9 interacts with OsEIN2, a central regulator of rice ethylene signaling, and the N-terminal domain directly binds to the *OsEBF1/2* mRNAs for translational inhibition, allowing accumulation of transcription factor OsEIL1 to activate the downstream signaling. RNA-IP seq and CLIP-seq analyses reveal that MHZ9 associates with hundreds of RNAs. Ribo-seq analysis indicates that MHZ9 is required for the regulation of ~90% of genes translationally affected by ethylene. Our study identifies a translational regulator MHZ9, which mediates translational regulation of genes in response to ethylene, facilitating stress adaptation and trait improvement in rice.

Ethylene is a versatile gaseous hormone that plays considerable roles in processes of plant growth, development and environmental adaptation[1–3]. Ethylene signaling pathway has been studied by using *Arabidopsis* and rice as model species, which represent dicotyledonous and monocotyledonous plants, respectively.

In *Arabidopsis*, ethylene is perceived by five ethylene receptors in the endoplasmic reticulum (ER)[1]. In the absence of ethylene, receptors associate with CONSTITUTIVE TRIPLE RESPONSE 1 (CTR1), which interacts with and phosphorylates ETHYLENE INSENSITIVE 2 (EIN2) to prevent its signaling[4–6]. The EIN2 protein undergoes rapid decay by two F-box proteins, EIN2 TARGETING PROTEIN 1 (ETP1) and ETP2 in *Arabidopsis*[7]. Downstream of EIN2, ETHYLENE INSENSITIVE 3 (EIN3)/EIN3-like (EILs) are key transcription factors in ethylene signaling

pathway and their protein levels are tightly controlled by two F-box proteins, EIN3 BINDING F-BOX 1/2 (EBF1/2)[8,9]. Upon ethylene perception, receptor configuration is likely altered and CTR1 is inactivated, leading to the dephosphorylation of EIN2 and cleavage of EIN2-CEND (EIN2-C)[4–6]. EIN2-C is further translocated into nucleus and interacts with EIN2 NUCLEAR-ASSOCIATED PROTEIN 1 (ENAP1) for regulation of EIN3/EIL1 through histone modification[1,10–13]. In addition, EIN2-C targets the poly U motif in *EBF1/2* mRNA 3′ untranslated region (UTR) for translational inhibition in RNA PROCESSING BODY (P-body), together with EIN5 and other P-body components[14,15]. Because EIN2 lacks distinct RNA-binding domains (RBDs), it raises the possibility that EIN2 may associate with *EBF1/2* mRNA bridged by an unidentified RNA-binding protein (RBP)[14]. Whether the RBPs are present and how they

[1]State Key Laboratory of Plant Genomics, Institute of Genetics and Developmental Biology, Innovative Academy for Seed Design, Chinese Academy of Sciences, Beijing 100101, China. [2]College of Advanced Agricultural Sciences, University of Chinese Academy of Sciences, Beijing 100049, China. [3]Guangdong Laboratory for Lingnan Modern Agriculture, College of Agriculture, South China Agricultural University, Guangzhou 510642, China. [4]State Key Laboratory of Crop Gene Exploration and Utilization in Southwest China, Sichuan Agricultural University, Wenjiang, Chengdu 611130, China. [5]MOA Key Laboratory of Pest Monitoring and Green Management, College of Plant Protection, China Agricultural University, Beijing 100193, China. [6]These authors contributed equally: Yi-Hua Huang, Jia-Qi Han, Biao Ma. ✉e-mail: chaoyang@cau.edu.cn; ccyin@genetics.ac.cn; jszhang@genetics.ac.cn

work with EIN2 to facilitate RNA binding and translational control remain unknown.

In rice, we identified several components of ethylene signaling through screening and analyses of a set of ethylene response mutants, *mao huzi* (*mhz*), which is a Chinese name of cat whiskers to describe the scattered adventitious roots in ethylene response mutants[16]. Compared with the ethylene signaling pathway in *Arabidopsis*, both conserved and diverged aspects have been found in rice. For conserved components, ethylene receptors OsERS1[3,17], OsERS2/MHZ12[3,18-20], OsETR2[18,21,22] and OsETR3[22], Raf-like kinase OsCTR1/2[23,24], membrane protein MHZ7/OsEIN2[16,25], and transcription factors MHZ6/OsEIL1 and OsEIL2[26,27] have been studied in rice. More regulators were discovered through mutant analysis in rice, i.e., a GDSL lipase MHZ11[24], a histidine kinase MHZ1[20] and an OsEIN2 stabilizer MHZ3[18]. Based on these studies, a rice ethylene signaling pathway has been proposed. In the absence of ethylene, OsERS2 interacts with both OsCTR2 and MHZ1 in the ER membrane. On the one hand, OsERS2 maintains the phosphorylation state of OsCTR2, which may then inhibit downstream OsEIN2-mediated signal transduction. On the other hand, OsERS2 represses the histidine kinase MHZ1 and prevents MHZ1-mediated phosphorelay process. With ethylene, the GDSL lipase MHZ11 reduces the sterol levels on the membrane to increase the membrane fluidity and flexibility, facilitating the ethylene-binding induced conformational change of OsERS2. Such conformational change would lead to the disassociation of OsERS2 with both OsCTR2 and MHZ1, resulting in the inactivation of OsCTR2 but activation of MHZ1[2,20,24]. Inactivation of the OsCTR2 may result in OsEIN2 stabilization by membrane protein MHZ3 for further downstream signaling events[18]. The activated MHZ1 would transfer the phosphoryl signal into the nucleus through a phosphorelay pathway, allowing for the activation of downstream signaling to regulate the root ethylene response in rice[2,20]. Both OsCTR2-OsEIN2 pathway and MHZ1-mediated pathway act downstream of the receptors to regulate ethylene responses in rice in parallel. While the OsCTR2-OsEIN2 pathway may function in whole plants, the MHZ1 pathway may only work in roots of rice[2,20].

Similar MHZ3 proteins, called MHZ3-LIKE1 (MHL1) and MHL2, have been found in Arabidopsis and may function as a positive regulator of ethylene responses[18]. Additional components such as MHZ4/ABA4[17], MHZ5/carotenoid isomerase[19], phospholipase GY1[28], E3 ubiquitin ligase MHZ2/SOR1[29] and tryptophan aminotransferase MHZ10/OsTAR2[30] are also found to function as cross-talk points with ABA, JA and auxin pathways for regulation of ethylene responses.

RBPs associate with RNAs forming ribonucleoprotein (RNP) complexes to regulate RNAs' metabolism and biological function ubiquitously, including biosynthesis, maturation, translation, and decay[31]. Some 3'UTRs of mRNAs have *cis*-elements that could be specifically recognized by RBPs to modulate their translation process[32]. Translational regulation is orchestrated by the combined actions of translation regulators (e.g., mammalian target of rapamycin, mTOR, and some RBPs)[33] and the regulatory elements present in the protein synthesis machinery. Beyond the defined translation machinery, the function of membrane-less organelles, termed RNA granules (e.g., P-body), has attracted increasing attention in modulating the mRNA translation process[34-36].

Here, through analysis of a rice ethylene-response mutant *mhz9*, we identified a glycine-tyrosine-phenylalanine (GYF) domain-containing protein MHZ9, which interacts with OsEIN2-C in P-body and directly binds to the *OsEBF1/2* mRNAs for translational repression. Such translational repression allows the accumulation of downstream transcription factors OsEILs and the activation of ethylene-responsive genes and the ethylene responses. Genes translationally regulated by MHZ9 in response to ethylene were also investigated at the whole-genome level. Our study identifies a master translational regulator MHZ9 and provides genome-wide insights into translational control of ethylene signaling in rice.

## Results

### Characterization of the *mhz9* mutant in ethylene responses

The *mhz9* mutant was isolated in our previous genetic screen for rice ethylene-response mutants[3,18] (Fig. 1). In ethylene, wild-type (WT) seedlings exhibited increased growth in coleoptiles but decreased growth in roots, while the *mhz9* mutant displayed insensitivity to ethylene in both root growth and coleoptile elongation (Fig. 1a). The induction of ethylene-responsive genes was abolished or reduced in *mhz9* (Fig. 1b). In addition, the field-grown plants of *mhz9* exhibit multiple phenotypic defects of agronomic traits (Supplementary Fig. 1).

Through mapped-based cloning, the *MHZ9* gene was mapped to *LOC_Os01g69990*. The loss of a single base A at 1045 bp of *MHZ9* coding sequence led to a premature stop codon in *mhz9* (Fig. 1c and Supplementary Fig. 2a). Both genetic complementation and RNAi experiments demonstrated that mutation of the *MHZ9* (*LOC_Os01g69990*) gene leads to ethylene response defect and field-grown phenotypes of *mhz9* (Supplementary Fig. 2b-d).

The *MHZ9* gene encodes a protein with a GYF domain, which is known for proline-rich peptide recognition[37] in animals. However, the *MHZ9* gene carrying mutations of each of the six signature residues in GYF domain still rescued the *mhz9* ethylene response (Supplementary Fig. 3), probably implying that these conserved sites may not work as that in animals. MHZ9 protein has homologs in different species (Supplementary Fig. 4). Phylogenetic analysis showed that MHZ9 clustered with those proteins from monocots and was distantly related with the homologs from Arabidopsis. Whether these homologs play similar roles in regulation of ethylene responses among different plant species needs further investigation.

The transcript level of *MHZ9* was mildly influenced by ethylene treatment (Supplementary Fig. 5a). Overexpression of *MHZ9* rendered enhanced ethylene responses and induction of ethylene-responsive genes in roots (Fig. 1d and Supplementary Fig. 5b), indicating that MHZ9 played a positive role in ethylene signaling. Ethylene perception inhibitor 1-MCP suppressed the constitutive short root phenotype of the *MHZ9*-OX seedlings (Fig. 1d). The *MHZ9* promoter was activated in coleoptiles and roots of young seedlings, and in leaf sheath, nodes and anthers and pistils of young flowers as revealed by GUS staining (Supplementary Fig. 5c). Under field conditions, overexpression of *MHZ9* led to a significant increase in plant height compared to WT (Supplementary Fig. 5d).

Together, all these results indicate that MHZ9 acts as a positive regulator of ethylene responses and affects multiple agronomic traits under field conditions.

### Genetic interactions between *MHZ9* and ethylene signaling components

Loss-of-function mutant *Osers2* showed mild ethylene hypersensitivity in roots, and this hypersensitivity was completely abolished in *Osers2 mhz9* double mutant, suggesting that *MHZ9* may function downstream of ethylene receptors (Fig. 2a). In our previous study, OsCTR2 phosphorylation status has been used as a biochemical indicator of ethylene response[24]. Ethylene reduced the phosphorylation status of OsCTR2 in both WT and *mhz9* mutant, implying that *MHZ9* mutation does not affect the OsCTR2 signal output (Supplementary Fig. 6a).

Overexpression of *MHZ9* resulted in a weak hypersensitivity in roots (Fig. 2b). However, seedlings with overexpression of *MHZ9* in *Osein2* background displayed a complete insensitivity to ethylene in both roots and coleoptiles (Fig. 2b). By contrast, the ethylene-induced hypersensitive response in *OsEIN2*-OX was attenuated in *OsEIN2*-OX/*mhz9* (Fig. 2b). The double mutant *Osein2 mhz9* showed complete ethylene insensitivity in both roots and coleoptiles growth (Fig. 2b). All the results indicate that both *MHZ9* and *OsEIN2* are required for each other's function in ethylene signaling.

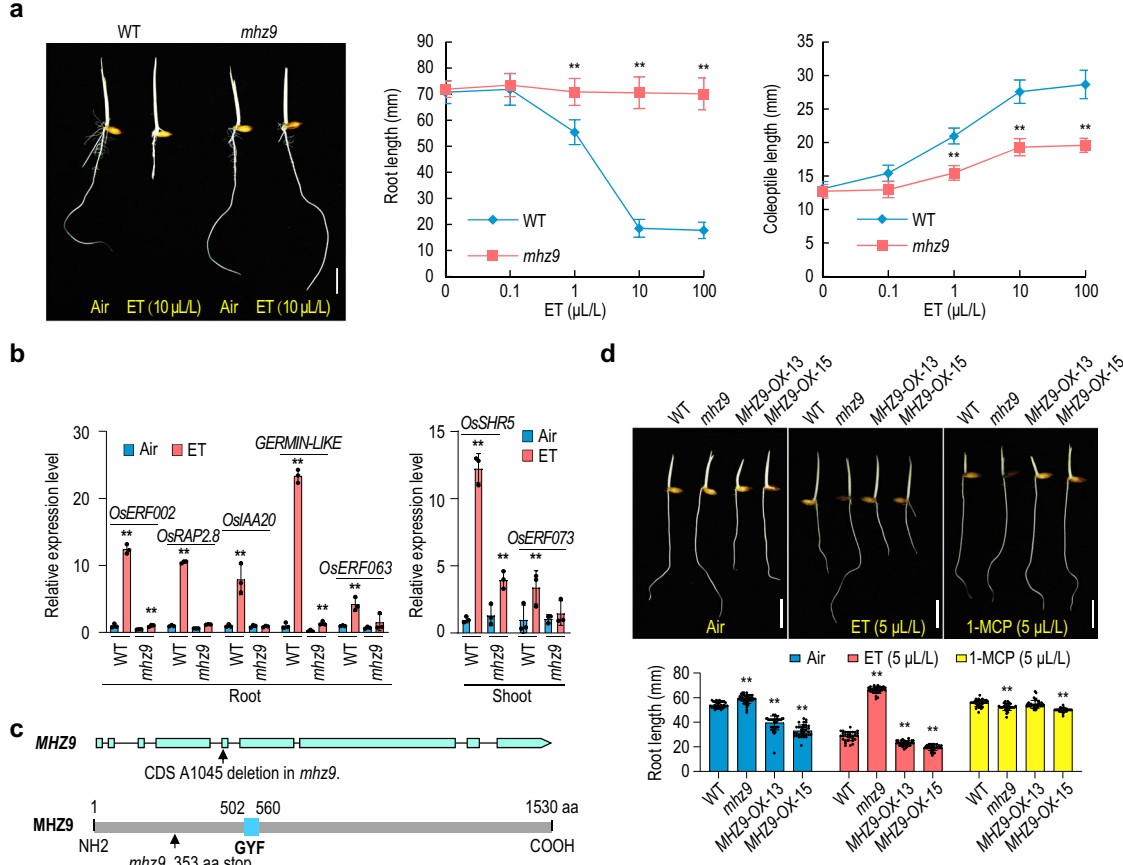

**Fig. 1 | Characterization and gene identification of *mhz9*. a** Ethylene-response phenotype of *mhz9*. Seedlings were grown in the dark for 3 d in the absence (air) or presence of 10 μL/L of ethylene (left). Ethylene dose-response curves for root length and coleoptile length of WT (Nipponbare) and *mhz9* (right). **b** Expression levels of ethylene-responsive genes in WT and *mhz9* with or without ethylene. Total RNAs were isolated from two-day-old etiolated seedlings treated with or without 10 μL/L of ethylene for 8 h. The relative RNA levels of ethylene-responsive genes were detected by RT-qPCR. *OsActin* was used for normalization. **c** Schematic

diagrams of *MHZ9* (*LOC_Os01g69990*) gene structure and MHZ9 domain. **d** Phenotype of *MHZ9*-overexpressing transgenic lines. **a**, **d** Lengths are means ± SD (*n* > 30 biologically independent samples). The scale bar indicates 10 mm. **b**, **d** The asterisks indicate significant differences compared with the corresponding WT controls (**$P < 0.01$; two-tailed Student's *t*-test). The experiments were repeated at least three times with similar results. And one representative set of results was shown. Source data are provided as a Source Data file.

*OsEIL1*-OX and *OsEIL2*-OX plants showed a hypersensitive response to ethylene in root inhibition and coleoptile promotion[27] (Fig. 2c), while the *OsEIL1/2*-OX/*mhz9* seedlings exhibited reduced ethylene response in roots and coleoptiles upon ethylene treatment (Fig. 2c). From all the genetic analyses, *MHZ9* may function downstream of *OsERS2*, and be required for *OsEIN2* and *OsEIL1/2* function in ethylene signaling, acting either upstream or downstream of *OsEIL1/2*.

**MHZ9 is localized in the P-body and interacts with OsEIN2-C**
The GFP-MHZ9 was localized in cytoplasmic dot-like structures, which aggregate further upon ethylene treatment (Fig. 3a). These dot-like structures of MHZ9 resembled the appearance of P-body that plays important roles in regulating ethylene signaling in previous studies in Arabidopsis[14,15]. An RFP-tagged OsEIN5, whose Arabidopsis homolog EIN5 is a P-body marker[38], was co-localized with MHZ9 in cytoplasmic foci (Fig. 3b). Previous study revealed that Arabidopsis EIN2-C and EIN5 could co-localize in P-body[14]. We found that MHZ9 co-localized with OsEIN2-C, and OsEIN2-C could co-localize with OsEIN5. The three proteins also co-localized in dot-like structures (Fig. 3b). Another P-body marker, decapping enzyme 2 (OsDCP2)[39] co-localized with MHZ9 (Fig. 3b). All these results support that MHZ9 is most likely a P-body localized protein.

The MHZ9 amino acid sequence was analyzed in detail and two additional putative domains were identified besides the GYF

domain (Fig. 3c). Through NCBI conserved domain search with the KOG v1.0-4825 PSSMS database, MHZ9-N terminus was predicated to contain a putative RNA splicing and modification domain PRP4 (72–233 aa), and through https://www.ebi.ac.uk/interpro/, MHZ9-C terminus was predicated to harbor a putative Q-rich region (866–1234) possibly associated with protein aggregation[40]. Co-localization of various MHZ9 versions (Fig. 3c) with P-body marker OsEIN5 was investigated, and the Q-3 (866–1020 aa) in MHZ9-Q-rich region alone is sufficient for MHZ9 localization in P-body. Deletion of the region of 909–975 aa in Q-rich region largely disrupted the protein aggregation in P-body (Fig. 3d).

To determine the molecular mechanism by which MHZ9 regulates ethylene signaling, we analyzed the potential proteins that interacted with MHZ9 through Immunoprecipitation-mass spectrometry (IP-MS). From the IP-MS analysis, one unique peptide corresponding to the end of the OsEIN2 C terminus was detected (Supplementary Data 1). The interaction between MHZ9 or its truncated versions and OsEIN2-C was further examined using yeast-two-hybrid, BiFC and Co-IP assays. MHZ9 was found to physically associate with OsEIN2-C in P-body through its Q-3 region (Fig. 3e–h). However, the P-body localization of MHZ9 and OsEIN2-C are independent of each other's presence (Supplementary Fig. 6b). Besides, the OsEIN2 abundance was only weakly influenced in *mhz9* compared to that in WT (Supplementary Fig. 6c).

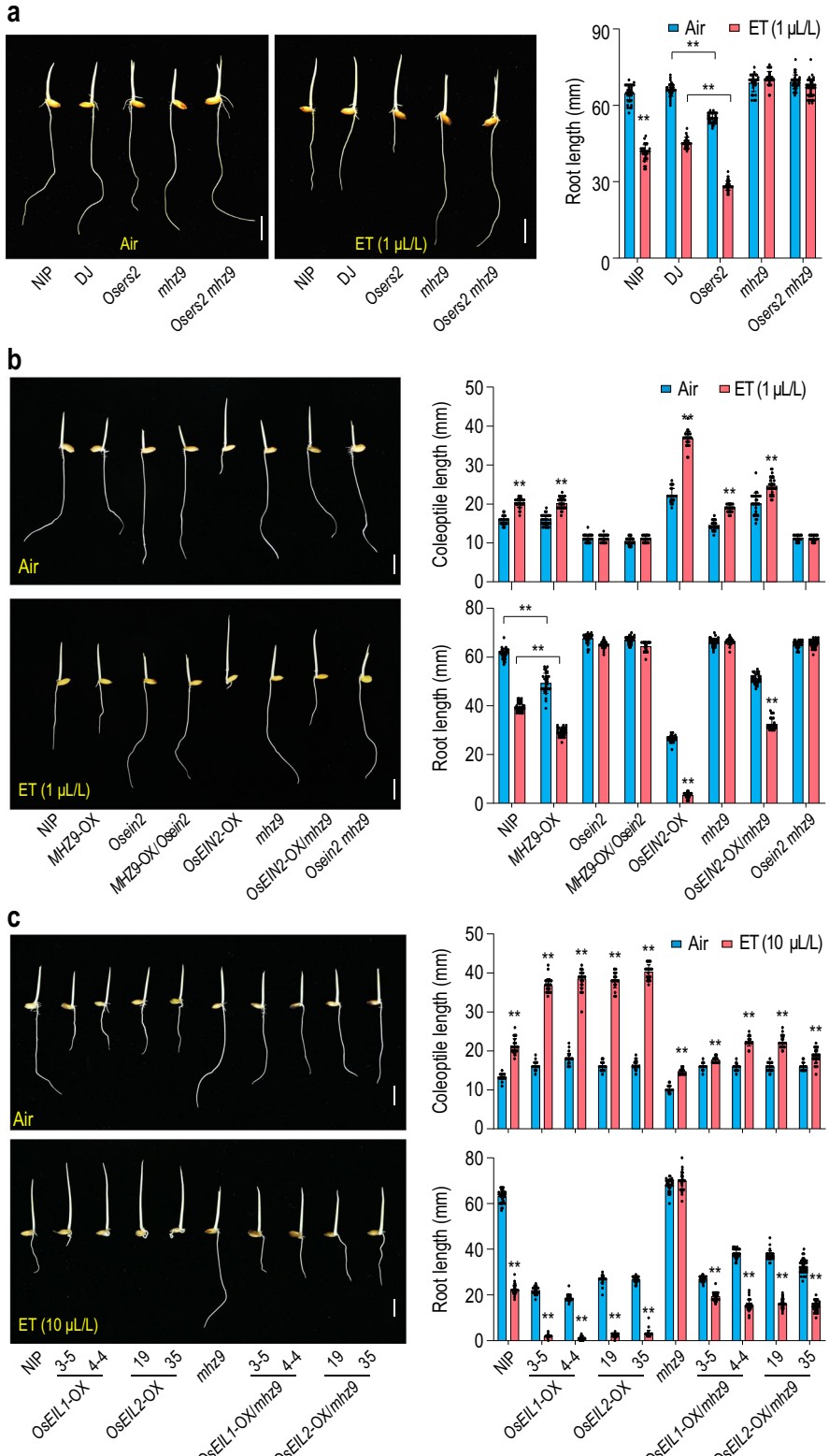

**Fig. 2 | Genetic interactions between *MHZ9* and ethylene signaling components. a** Ethylene-response of *Osers2*, *mhz9* and *Osers2 mhz9* double mutants. *Osers2* is in the DJ background, and *mhz9* is in WT (Nipponbare) background. **b** Genetic interaction between *MHZ9* and *OsEIN2*. **c** Ethylene-response of *OsEIL1*- or *OsEIL2*-overexpression etiolated seedlings in WT and *mhz9* background. **a**–**c** Lengths are means ± SD (*n* > 30 biologically independent samples). The asterisks indicate significant differences compared with the corresponding controls (**\**P* < 0.01; two-tailed Student's *t*-test). The scale bar indicates 10 mm. The experiments were repeated at least three times with similar results, and one representative set of results was shown. Source data are provided as a Source Data file.

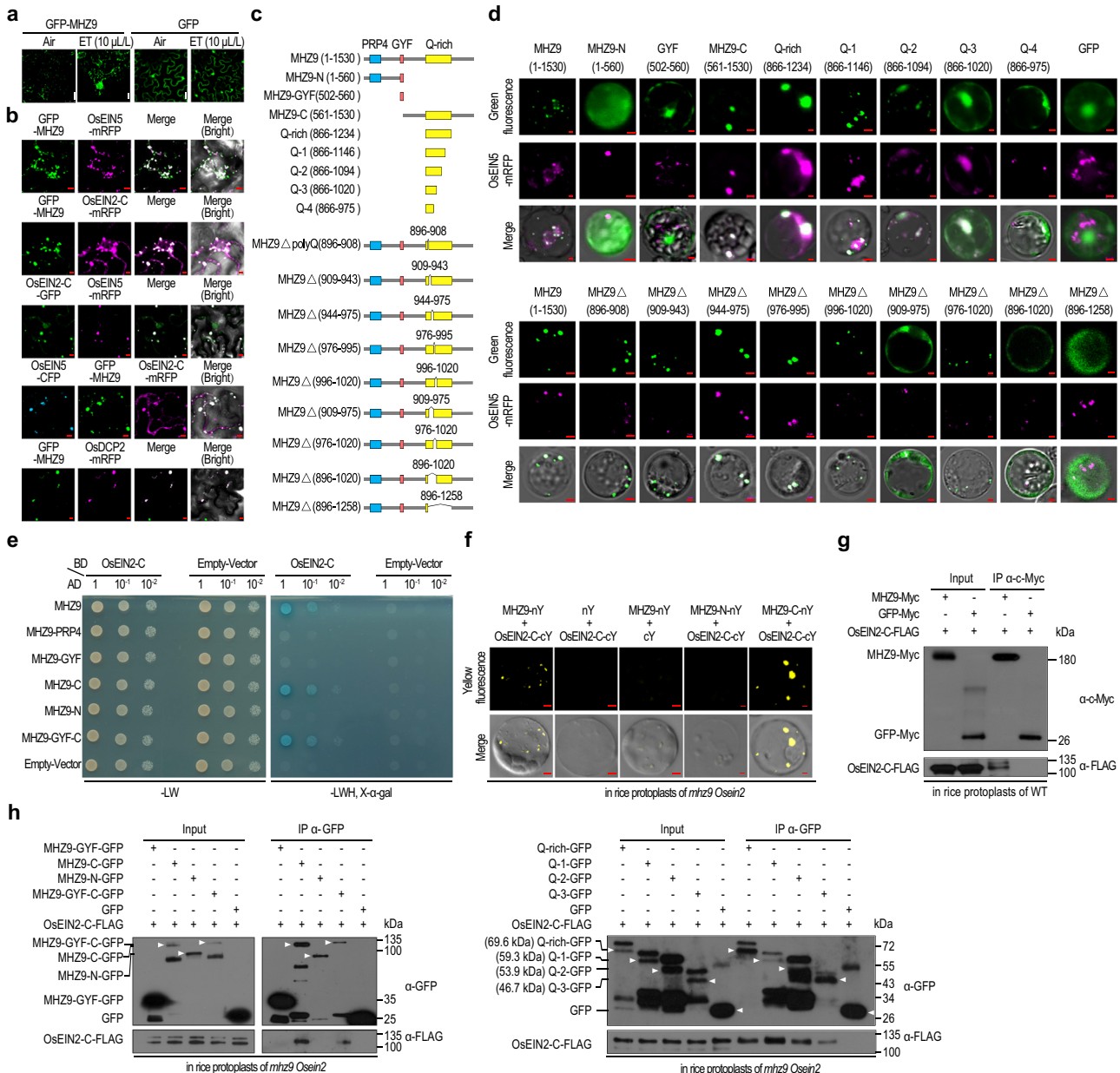

**Fig. 3 | MHZ9 is localized in P-body and interacts with OsEIN2. a** Subcellular localization of MHZ9. GFP-MHZ9 was transiently expressed in tobacco leaves by *Agrobacterium* infiltration and treated with or without 10 μL/L of ethylene for 16 h at 2 days post infiltration. GFP was used as the control. Scale bars, 20 μm. **b** The co-localization tests of MHZ9, OsEIN5 (P-body marker), OsEIN2-C, and OsDCP2 (P-body marker). The co-localization assays of MHZ9 with OsEIN5 (the first row), MHZ9 with OsEIN2-C (the second row), OsEIN2-C with OsEIN5 (the third row), the three proteins (OsEIN5, MHZ9, and OsEIN2-C), and MHZ9 with OsDCP2 were performed by using *Agrobacterium*-mediated transient expression in tobacco leaves followed by treatment of 10 μL/L of ethylene for 16 h at 2 d post infiltration. Scale bars are 5 μm. **c** Diagrams of full-length and truncated versions of MHZ9 used in subcellular localization and interaction domain mapping studies. **d** The co-localization study of truncation (upper panel) and deletion (lower panel) variants of MHZ9 with OsEIN5 in *mhz9* protoplasts. Scale bars, 2 μm. **e** Yeast-two-hybrid assay

for interaction between MHZ9 and OsEIN2-C. Full-length and truncated versions of MHZ9 used are the same as in (**c**). The region for MHZ9-PRP4 is 72–233 aa and the region for MHZ9-GYF-C is 502–1530 aa. **f** BiFC assays for interaction between different truncated versions of MHZ9 and OsEIN2-C. Scale bar, 10 μm. Interaction analysis of full-length of MHZ9 with OsEIN2-C (**g**), and different MHZ9 domains with OsEIN2-C (**h**) by co-IP assays. The total proteins were immunoprecipitated with anti-Myc affinity gel (**g**) or GFP-Trap (**h**), GFP-Myc (**g**) or GFP (**h**) was used as the negative control. The estimated molecular weight of each protein in (right panel of **h**) was indicated in parentheses. The band corresponding to each of the truncated proteins is marked by a triangle, and the upper bands with higher molecular weights than expected may arise from protein modifications. Each experiment was repeated at least three times with similar results. Source data are provided as a Source Data file.

## The N-terminal domain of MHZ9 binds to *OsEBF1/2* mRNAs

MHZ9-N-terminal end was predicted to have a PRP4 domain possibly involved in RNA binding. Through XRNAX-IP-MS method[41], we identified RNA base-conjugated peptides in the middle region of MHZ9-N (Fig. 4a). We further performed RNA-IP (RIP)-seq and identified 1777 genes possibly bound by MHZ9-N (Fig. 4b, Supplementary Data 2). The

binding sites of MHZ9 on its target RNAs were also studied through CLIP-seq analysis. The RNAs identified from CLIP-seq data (Supplementary Data 3) were compared with those from the RNA-IP seq data. Totally 626 genes were overlapped and these genes were regarded as the most stringent MHZ9 targets (Fig. 4b, Supplementary Data 4). From these 626 genes, 1186 MHZ9-binding CLIP sites were identified,

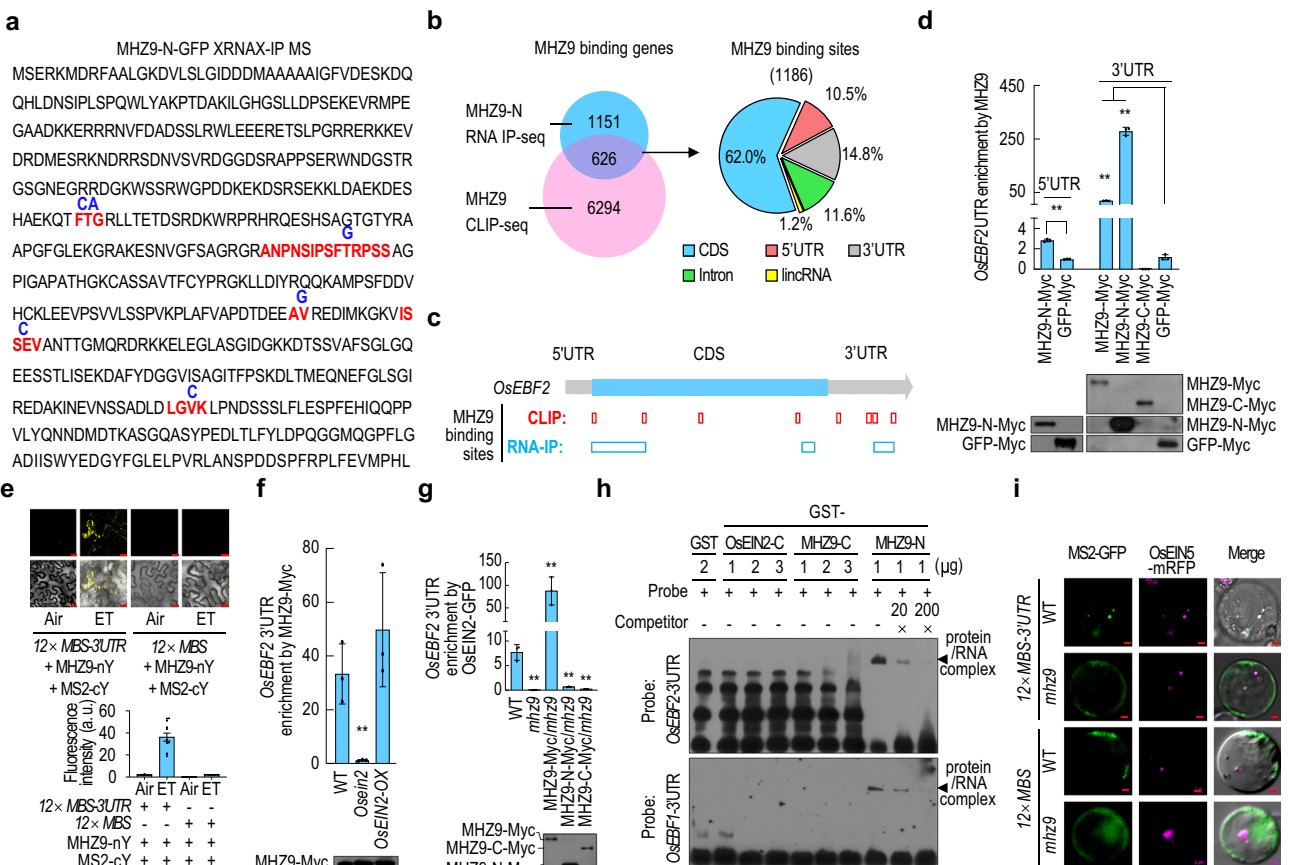

**Fig. 4 | MHZ9-N-terminal domain binds to the 3′UTR of *OsEBF1/2* mRNA. a** The RNA bases identified from purified MHZ9-N peptides are indicated in dark blue. The RNA-bound peptides are highlighted in red. **b** MHZ9-binding genes and binding sites detected through RNA-IP (RIP)-seq and crosslinking and immunoprecipitation followed by sequencing (CLIP-seq). Then the 1186 MHZ9-binding sites identified from the overlapped 626 genes (left) were analyzed to investigate their distribution patterns (right). **c** Schematic representation of the MHZ9-binding sites on *OsEBF2* mRNA from RIP-seq and CLIP-seq. **d** Binding test of *OsEBF2* mRNA UTRs by various MHZ9 truncations through RIP-qPCR assay. The protein expression levels were also examined. **e** Binding tests of MHZ9 to *OsEBF2* mRNA 3′UTR through MS2-based BiFC assay. *12 × MS2 binding sites* (*12 × MBS*) fused with or without *OsEBF2* mRNA 3′UTR was transiently expressed with MS2-cYFP and MHZ9-nYFP in tobacco leaves and treated with 10 µL/L of ethylene or air for 16 h at 2 days post infiltration. Scale

bars, 20 µm. Values are means ± SD (*n* = 12 biologically independent samples). **f** OsEIN2 effect on the binding of MHZ9 to *OsEBF2* mRNA 3′UTR through RIP-qPCR. **g** MHZ9 effect on the association between OsEIN2 and *OsEBF2* mRNA 3′UTR through RIP-qPCR. **d**, **f**, **g** Values are means ± SD (*n* = 3 biologically independent samples). The asterisks indicate significant differences compared with the corresponding control (**P < 0.01; two-tailed Student's *t*-test). **h** MHZ9-N binds to the 3′UTR of *OsEBF1/2* mRNA directly in an RNA-EMSA assay. Arrows and braces indicate shifted protein/RNA complex bands. **i** Roles of MHZ9 in co-localization of *OsEBF2* mRNA 3′UTR and OsEIN5-RFP in P-body. *12 × MS2 binding sites* (*12 × MBS*) fused with or without *OsEBF2* mRNA 3′UTR was transiently expressed with MS2-GFP and OsEIN5-RFP in rice protoplasts of WT or *mhz9* mutant. Scale bars, 2 µm. Each experiment was repeated at least three times with similar results. Source data are provided as a Source Data file.

and a large portion (62%) is located at the coding region of mRNAs (Fig. 4b). About 10.5% and 14.8% of the binding sites are located at 5′UTR and 3′UTR respectively. The remaining sites are present at introns and long intergenic non-coding RNAs (lincRNAs) (Fig. 4b). Gene Ontology (GO) enrichment analysis revealed that the 626 MHZ9-binding targets are likely involved in responses to water deprivation, hormone, abiotic stress and other stimuli (Supplementary Fig. 7a). The potential consensus motifs and secondary structures shared by MHZ9-binding targets were also analyzed, and UC/AG-rich elements and stem-loop structures were primarily enriched (Supplementary Fig. 7b, c).

Among the 626 MHZ9-associated RNA targets, MHZ9-binding sites on the mRNA of *OsEBF2*, encoding a homolog of Arabidopsis EBF1/2 that mediates degradation of transcription factors EIN3/EIL1 in ethylene signaling pathway[8,9], were highly enriched (Fig. 4c, Supplementary Data 2, 3). Because the 3′UTR and fragments near 5′UTR of *OsEBF2* mRNA were enriched in MHZ9-N RIP-seq (Fig. 4c, Supplementary Data 2), the 35S::5′UTR−6 × HIS and 35S::6 × HIS−3′UTR constructs were made and transfected into the *mhz9* protoplasts, together with the constructs expressing different versions of MHZ9, for RIP-

qPCR analysis. MHZ9-N showed a much higher binding ability to 3′UTR of *OsEBF2* mRNA than that to 5′UTR in comparison to the corresponding GFP controls (Fig. 4d). The full length of MHZ9 also exhibited a high binding ability to the 3′UTR. In contrast, MHZ9-C did not show significant binding to the 3′UTR (Fig. 4d). The results clearly show that MHZ9-N containing the PRP4 domain can bind to the *OsEBF2* 3′UTR.

The MS2 system is usually used for RNA visualization in living cells, in which the bacteriophage MS2 coat protein (MCP) is fused to a fluorescent protein and binds to a specific RNA hairpin sequence MBS (MS2-binding site), allowing for the visualization of MBS-contained RNAs[42]. We designed and performed the MS2[14,43]-based BiFC assays to detect the ethylene effect on binding of MHZ9 to 3′UTR of *OsEBF2* mRNA. After ethylene treatment, a strong YFP signal was detected in cytoplasmic dot-like structures when MS2-cYFP, MHZ9-nYFP and *12 × MBS*-3′UTR were co-expressed, while little or no yellow florescence was detected in air or control (Fig. 4e), indicating that the MHZ9 protein binds to the 3′UTR of *OsEBF2* mRNA. Since MHZ9 interacts with OsEIN2 in P-body (Fig. 3e–h), we investigated whether OsEIN2 could affect MHZ9 association with *OsEBF2* 3′UTR. RIP-qPCR assays revealed that the binding of MHZ9 to *OsEBF2* 3′UTR was largely abolished in *Osein2*

protoplasts compared with WT, while *OsEIN2* overexpression mildly promoted the binding (Fig. 4f).

We also investigated whether OsEIN2 could associate with *OsEBF2* 3′UTR. The results showed that the OsEIN2 protein associated with the *OsEBF2* 3′UTR in WT protoplasts but hardly in *mhz9* protoplasts (Fig. 4g). Only MHZ9 full-length but not MHZ9-N or MHZ9-C could restore the association between OsEIN2 and *OsEBF2* 3′UTR (Fig. 4g). Taken together, the intact MHZ9 and OsEIN2 are mutually required for association with *OsEBF2* 3′UTR in plant cells.

To determine whether MHZ9 directly binds to the *OsEBF1/2* mRNA 3′UTR, we performed an RNA- electrophoretic mobility shift assay (EMSA). The GST-MHZ9-N could bind to the probe of *OsEBF2* mRNA 3′UTR in a dose dependent manner as revealed from the protein-RNA complex, while no detectable bands were observed for GST-OsEIN2-C, GST-MHZ9-C or the GST control (Supplementary Fig. 8, Fig. 4h). Similarly, the GST-MHZ9-N, but not the GST-MHZ9-C, GST-OsEIN2-C or the GST control, could also directly bind to the probe of *OsEBF1* mRNA 3′UTR mildly (Supplementary Fig. 8, Fig. 4h).

The roles of MHZ9 in possible subcellular localization of *OsEBF2* mRNA 3′UTR were examined using the MS2 system. In WT, *OsEBF2* mRNA 3′UTR indicated by MS2-GFP could co-localize with OsEIN5-RFP in cytoplasmic foci, while in *mhz9*, such co-localization was disrupted (Fig. 4i), suggesting that the 3′UTR-mediated P-body localization of the *OsEBF2* mRNA requires MHZ9 function.

## MHZ9 represses *OsEBF1/2* mRNA translation

Since MHZ9 directly binds to the *OsEBF1/2* mRNA and is localized in P-body, and P-body is usually involved in RNA degradation and translational control[44,45], we examined the *OsEBF1/2* mRNA stability and expression level in WT and *mhz9* mutant. *MHZ9* mutation significantly reduced the *OsEBF1/2* mRNA stability and abundance (Supplementary Fig. 9a). However, considering that OsEBF1 and OsEBF2 play a negative role in ethylene signaling, the less *OsEBF1/2* mRNA stability and abundance in *mhz9* mutant appears not to affect the ethylene insensitivity of the *mhz9* mutant. We next analyzed the influence of MHZ9 on translation efficiency (TE) of *OsEBF1/2* mRNA during ethylene response through polysome profiling (Fig. 5a). In WT, ethylene reduced the abundance of *OsEBF1/2* mRNA in fractions 12–14 containing relatively large polysomes, and this reduction was strengthened with longer time of ethylene treatment (Fig. 5b). In contrast, ethylene treatment only led to slight changes in *mhz9* mutant (Fig. 5b). The translation states of the putative reference mRNAs (*OsEF1α*, *OsUBQ5* and *OsGADPH*) were also examined and they were less affected by ethylene in both WT and *mhz9* mutant. However, the translation levels of these reference mRNAs were apparently reduced in the *mhz9* mutant (Fig. 5b). These results indicate that ethylene-provoked translation inhibition on *OsEBF1/2* mRNA is dependent on MHZ9. Other genes may also be affected by MHZ9 independent of ethylene function.

We further evaluated roles of the 3′UTRs of *OsEBF1/2* mRNA in translational inhibition by MHZ9 and OsEIN2 using an optimized dual luciferase assay. The plasmid harboring both the reference *Renilla* luciferase (*R-Luc*) gene and the reporter *Firefly* luciferase gene (*F-LUC*) - 3′UTR of *OsEBF1/2* was transfected into protoplasts. In WT protoplasts, the LUC activity was reduced to about 70% and 26% when 3′UTR of *OsEBF1* and *OsEBF2* was included respectively (Supplementary Fig. 9b). In *mhz9* protoplasts, the reduction of LUC activity mediated by 3′UTR of *OsEBF2* is largely abolished while the activity of 3′UTR of *OsEBF1* is comparable to that in WT protoplasts. In *Osein2* protoplasts, the LUC activity was further recovered for both *OsEBF1* and *OsEBF2* 3′UTRs (Supplementary Fig. 9b). The RNA levels of various *LUC* fusions were reduced for 3′UTR fused genes in *mhz9* and *Osein2* protoplasts (Supplementary Fig. 9b). Endogenous *OsEBF1/2* mRNA levels in *mhz9* and *Osein2* mutants were also much less than those in WT (Supplementary Fig. 9c). The TEs of *LUC-3′UTR* of *OsEBF1/2* were then calculated by the ratio of LUC activity versus its RNA levels, and these TE values were significantly reduced in WT protoplasts compared with the *LUC* controls. However, such inhibitory effects were abolished in *mhz9* and *Osein2* protoplasts, and in *Osein2* protoplasts, the activity was even much higher than the controls (Fig. 5c). These results indicate that the 3′UTR of *OsEBF1/2* mediates translational inhibition and the inhibition is dependent on both MHZ9 and OsEIN2. Additional factors may be involved in OsEIN2-mediated translational inhibition on *OsEBF2*.

We next detected the OsEBF2 protein levels to study the roles of 3′UTR in translational regulation and the function of MHZ9 and OsEIN2 in this process. The OsEBF2 protein accumulated at a higher level in *mhz9* and *Osein2* protoplasts than that in WT protoplasts, when the 3′UTR was included in *GFP-OsEBF2* (Fig. 5d). In WT protoplasts, when the 3′UTR was removed from the *GFP-OsEBF2*, the OsEBF2 level is higher than that from the fusion gene with 3′UTR. In *Osein2* protoplasts, the OsEBF2 level was very similar when 3′UTR was included in or excluded from the gene (Fig. 5d). In *mhz9* protoplasts, the OsEBF2 level seemed to resemble the case in *Osein2* protoplasts and even accumulated to a slightly higher level. It is possible that other OsEIN2-independent pathways may exist to repress *OsEBF1/2* mRNA expression through MHZ9, and MHZ9-binding sites in the coding region of *OsEBF1/2* mRNA may also contribute to this regulation (Fig. 4c). All the results indicate that MHZ9 and OsEIN2 could inhibit *OsEBF2* translation through 3′UTR function of *OsEBF2* mRNA.

Given that EBF1 and EBF2 are responsible for the degradation of EIN3/EIL1 in both Arabidopsis and rice[8,9,46], the MHZ9 effect on OsEIL1 accumulation was detected. The OsEIL1 protein was apparently induced and steadily accumulated after ethylene treatment in WT background in both shoots and roots (Supplementary Fig. 9d and Fig. 5e). In contrast, in the *mhz9* mutant, the OsEIL1 protein was only slightly induced and maintained at a much lower level (Supplementary Fig. 9d and Fig. 5e). Considering that the ethylene-exerted RNA profile of *OsEIL1-FLAG* was only slightly affected in *mhz9* mutant compared to that in WT (Supplementary Fig. 9e), we propose that MHZ9 is required for the OsEIL1 protein accumulation most likely through MHZ9 suppression of *OsEBF2* mRNA translation, diminishing OsEBF2-mediated proteasomal degradation of OsEIL1.

We explored whether MHZ9 could directly inhibit *OsEBF2* mRNA translation in in vitro translation system. In the presence of MBP-MHZ9, the GFP-OsEBF2 protein level from the *GFP-OsEBF2* mRNA with 3′UTR was reduced compared to the effect of control MBP. In contrast, MBP-MHZ9 exerted no significant reduction in GFP-OsEBF2 level when *GFP-OsEBF2* mRNA without 3′UTR was used compared to the MBP effect (Fig. 5f).

Epistasis analysis of *MHZ9* and *OsEBF1/2* was further performed. Both *Osebf1* and *Osebf2* single loss-of-function mutants exhibited constitutive short root phenotype in air and enhanced ethylene-response phenotype (Fig. 5g). The *Osebf1 Osebf2* loss-of-function mutant phenotype was more severe than the single loss-of-function mutants (Fig. 5g). Mutation of *OsEBF1*, *OsEBF2* or both in *mhz9* also led to short roots compared to WT in air, suggesting that *MHZ9* may function at or upstream of *OsEBF1/2* for root regulation in air. In ethylene, the *Osebf1 mhz9*, *Osebf2 mhz9*, and *Osebf1 Osebf2 mhz9* loss-of-function mutants all displayed partially insensitive responses to ethylene in both roots and coleoptiles compared to the *Osebf1*, *Osebf2* or *Osebf1 Osebf2* loss-of-function mutants (Fig. 5g), indicating that *MHZ9* is required for the ethylene-response phenotype of *Osebf1/2* loss-of-function mutants. These results also suggest that MHZ9 may have additional regulatory effects on ethylene response independent of the *OsEBF1/2* function.

## Analyzing the effects of MHZ9 on global translational control

Besides *OsEBF1/2* mRNA, we sought to investigate whether MHZ9 can exert translational regulation at the whole-genome level during ethylene response. We performed ribosome-footprint analysis (Ribo-seq)

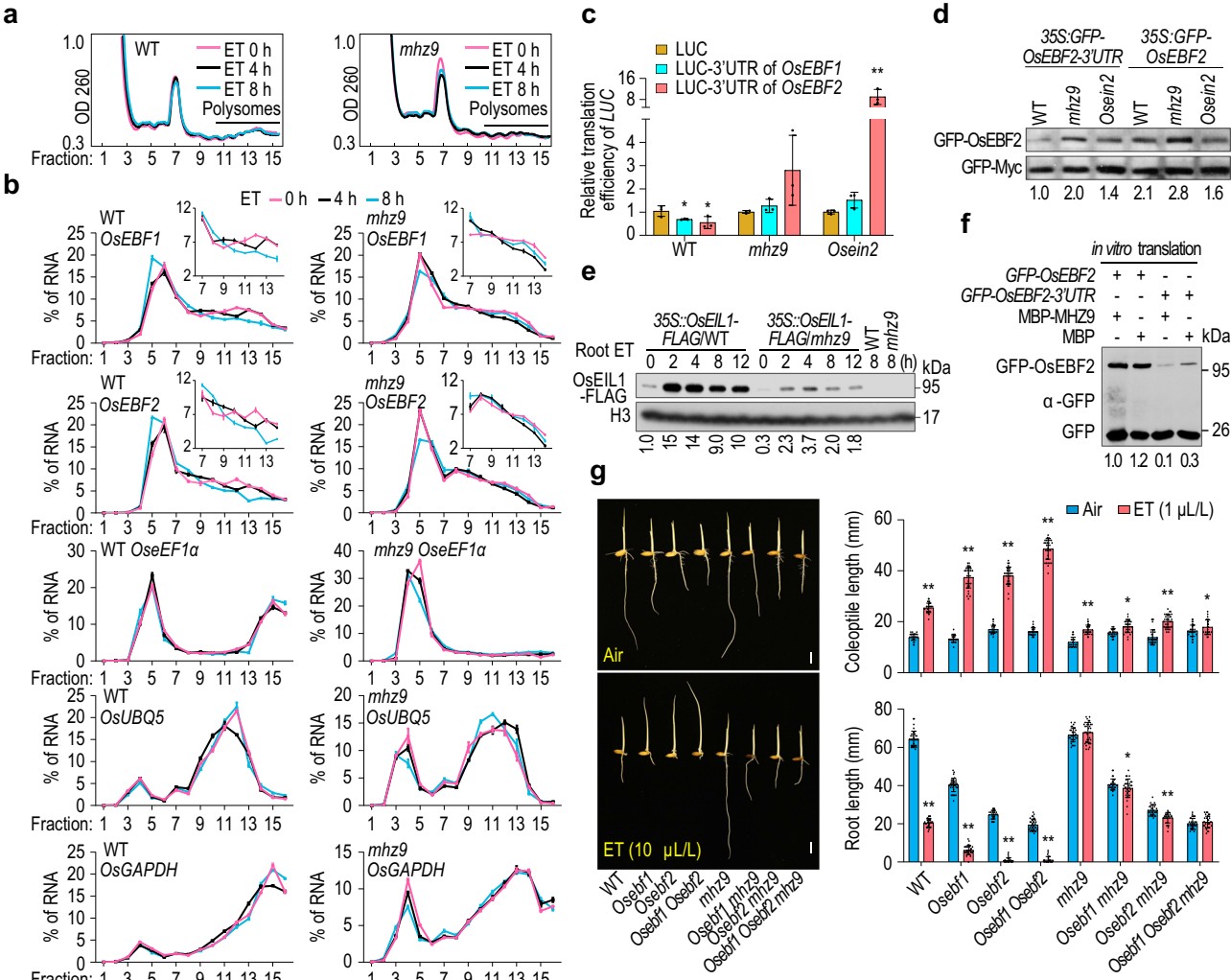

**Fig. 5 | MHZ9 mutation facilitates the translation of OsEBF1/2 mRNA. a** Polysome profiling assays with etiolated rice seedlings of WT and *mhz9*. The two-day-old etiolated seedlings were treated with ethylene for 0, 4, or 8 h, and subjected to polysome profiling. **b** Comparison of the relative levels of *OsEBF1*, *OsEBF2*, *OseEF1α*, *OsUBQ5* and *OsGAPDH* mRNAs in each gradient fraction in WT and *mhz9* mutant. The mRNA level in each fraction was detected through RT-qPCR and quantified with the raw CT value as a percentage relative to the total. *OseEF1α*, *OsUBQ5*, and *OsGAPDH* were used as the reference gene. The values are means ± SD (*n* = 3). **c** Roles of MHZ9 and OsEIN2 in translational inhibition mediated by *3'UTR* of *OsEBF1* or *2*. Relative TEs of *LUC* were calculated by the ratio of LUC activity to its mRNA levels. The values are means ± SD (*n* = 3 biologically independent samples), and the asterisks indicate significant differences compared with the corresponding controls (**P < 0.05; one-tailed Student's *t*-test for WT test. **P < 0.01; two-tailed

Student's *t*-test for *Osein2* test). **d** Comparison of OsEBF2 protein levels in WT, *mhz9*, and *Osein2* mutant. The constructs were co-expressed in WT, *mhz9* and *Osein2* protoplasts. GFP-Myc served as an internal control. **e** Accumulation of OsEIL1 in *35 S::OsEIL1-FLAG*/WT and *35S::OsEIL1-FLAG*/*mhz9* plants in response to ethylene. **f** MHZ9 effect on in vitro translation of *GFP-OsEBF2* mRNA fused with 3'UTR of *OsEBF2*. One µg of MBP and MBP-MHZ9 proteins were applied for in vitro translation tests, respectively. **g** Epistasis analysis of *MHZ9* and *OsEBF1/2*. Single, double, and triple mutants were generated through Crispr-Cas9 and/or crossing. Lengths are means ± SD (*n* > 30 biologically independent samples). The asterisks indicate significant differences compared with the corresponding controls (*P < 0.05; **P < 0.01; two-tailed Student's *t*-test). Data are representative of three independent experiments. Source data are provided as a Source Data file.

with WT and *mhz9* in the presence or absence of ethylene. The quality assessment showed a high degree of reproducibility between replicates, with adequate enrichment (over 90%) in the open reading frame (ORF), an obvious pause signal of ribosome around the start and stop codons, and the 3 nt periodicity (Supplementary Data 5 and 6, Supplementary Fig. 10).

The RNA-seq was also performed in parallel with the Ribo-seq to evaluate the mRNA level changes during ethylene response. After ethylene treatment for 4 h, 2717 (Fig. 6a, Supplementary Data 7) and 3511 (Fig. 6b, Supplementary Data 8) genes were significantly and differentially expressed at the transcriptional and translational levels, respectively. Translation efficiency (TE) analysis with the Ribo-seq reads/RNA-seq reads for each gene revealed that 1956 genes were significantly and differentially regulated at the TE level in response to

ethylene (Fig. 6c, Supplementary Data 9). These results illustrate the profound effects of ethylene on gene expressions at both the transcriptional and translational levels.

We then performed a comprehensive gene expression analysis in *mhz9* mutant in comparison with that in WT (Fig. 6d). Among the 2717 ethylene-induced differentially expressed genes in WT, *MHZ9* mutation blocked the transcriptional response of 1886 (~70%) genes. Among the 3511 translationally-altered and 1956 TE-altered genes in WT, patterns of 3279 (~93%) genes and 1788 (~91%) genes including *OsEBF1* were disrupted in *mhz9* mutant, respectively (Fig. 6d and Supplementary Data 7–9).

We further analyzed the distribution of the $\log_2$-TE ratio (ET/Air) of the 1956 TE-altered genes in WT and *mhz9* mutant. Upon ethylene treatment, a large variation in the $\log_2$-TE ratio

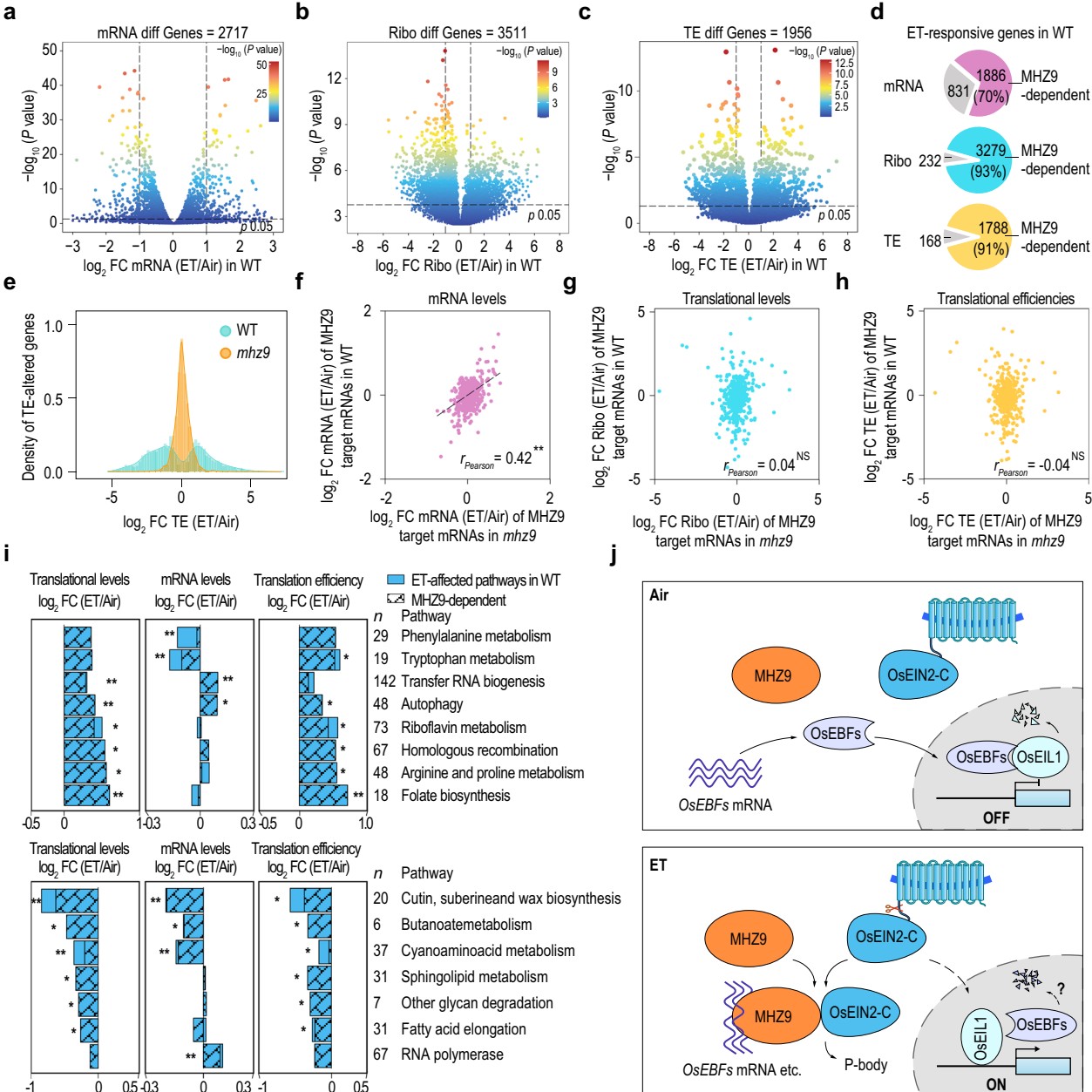

**Fig. 6 | Translational regulation of ethylene signaling is dependent on MHZ9.** Analysis of ethylene-induced alterations at **a** RNA, **b** translational, and **c** translation efficiency (TE) levels in WT. Two-day-old etiolated seedlings were treated with or without 10 μL/L of ethylene for 4 h and then were subjected to RNA-seq and ribosome footprints (Ribo-seq) analysis with two bioreplicates, respectively. The genes with RPM more than 0 were selected for differential genes identification. The black dashed line indicates the 0.05 $P$ value cutoff. **d** Comparison of the ethylene-responsive genes at mRNA (top), translational (middle), and TE (bottom) levels in WT and *mhz9*. The significantly differentially expressed genes in WT were compared with that in *mhz9* using Venn diagrams (Venny 2.0). **e** Distribution of $\log_2$ fold-changes of the 1956 TE-differential genes in WT and *mhz9*. The alterations of mRNA (**f**), translational (**g**), and TE (**h**) levels of the 555 MHZ9-binding genes (with both RNA-seq and Ribo-seq read > 0 RPM) in WT and *mhz9* during ethylene response. **f**–**h** **$P < 0.01$, NS, not significant. **i** KEGG analysis at the translational,

mRNA, and translation efficiency (TE) levels in response to ethylene. The $\log_2$FC (ET/Air) of each pathway was the average of $\log_2$FC of its contained genes. The pathways with statistically significant differences compared to the average of all pathway genes were selected and divided into up-regulated (top) and down-regulated (bottom) based on Ribo-seq (*$P < 0.05$, **$P < 0.01$; Mann–Whitney $U$ test). The number of $n$ indicates gene numbers contained in each pathway. **j** A proposed MHZ9 working model. In the absence of ethylene, MHZ9 remains at low-affinity-binding states to *OsEBFs* mRNA. OsEBFs proteins accumulate and lead to the degradation of OsEIL1. Ethylene-response is OFF. With ethylene, MHZ9 possibly receives a signal from OsEIN2-C through direct interaction, and then binds to the 3′ UTR of *OsEBFs* mRNA and other targets for translational inhibition, which contributes to the OsEIL1 protein accumulation and downstream ethylene signaling. The residual OsEBF1/2 may decay through unknown mechanisms.

(ET/Air) is observed in WT for TE-altered genes. However, the $\log_2$-TE ratio remained almost unchanged for most of these genes in *mhz9* (Fig. 6e). Together, all these results indicate that MHZ9 is a key regulator for the ethylene-induced translational response in rice.

MHZ9 mediates the translational response of 91% (1788) of TE-altered genes induced by ethylene (Fig. 6d). Among these genes, 105 (~6%) genes (Supplementary Data 10) were directly bound by MHZ9 through comparison with the 626 MHZ9-binding genes derived from the CLIP-seq and RIP-seq (Fig. 4b, Supplementary Data 4). It is

therefore possible that many of the MHZ9-dependent and TE-altered genes may be indirectly modulated by MHZ9 through OsEIL1/2, and/or other MHZ9-binding genes or via other mechanisms.

We further investigated the alterations of mRNA abundance and translational levels of 555 genes (RPM > 0 in both RNA-seq and Ribo-seq), including the above mentioned 105 genes, from the 626 MHZ9-binding genes (Fig. 4b, Supplementary Data 4). Alterations of mRNA abundance in *mhz9* were similar to those in WT as judged from the $r_{Pearson}$, suggesting that MHZ9 mutation may not significantly affect expression of its binding genes (Fig. 6f, Supplementary Data 10). However, the regulatory patterns of MHZ9-binding genes at both translational and TE levels were substantially disrupted in *mhz9* compared with those in WT, as revealed by the corresponding $r_{Pearson}$ values (Fig. 6g, h; Supplementary Data 10). These results indicate that MHZ9 may affect the translational response of most of its bound mRNAs in ethylene, with some genes being significantly altered.

Ethylene induced alterations of many genes at translational and TE levels (Fig. 6a–c), and KEGG pathway analysis was performed to find what pathways are involved in ethylene responses (Fig. 6i, Supplementary Data 11). The translational and TE promotions of phenylalanine and tryptophan metabolism, transfer RNA biogenesis, and autophagy depend on MHZ9 in ethylene responses (Fig. 6i). Among the MHZ9 down-regulated pathways, lipid metabolism was noted (Fig. 6i). These results indicate that MHZ9 may achieve its function through translational control of the above processes during ethylene responses in rice.

## Discussion

In this study, through analysis of a rice ethylene-response mutant *mhz9*, we have identified the GYF domain protein MHZ9, which binds to the *OsEBF1/2* mRNA for translational control in the ethylene response of rice. We propose that the C-terminal domain of MHZ9 may interact with OsEIN2-C to receive upstream ethylene signal and then directly bind to the 3′UTR and/or other regions of *OsEBF1/2* mRNA through MHZ9-N for translational inhibition in P-body. Suppression of *OsEBF2* translation would reduce OsEBF2-mediated degradation of the transcription factor OsEIL1, allowing accumulation of OsEIL1 for further activation of downstream signaling events. The residual OsEBF1/2 proteins may decay through unknown mechanisms (Fig. 6j). Genome-wide Ribo-seq, RIP-seq, and CLIP-seq analyses reveal that MHZ9 directly binds to 626 genes and, of these, MHZ9 significantly regulates the translation efficiencies (TEs) of 105 genes. Among the 1956 genes with altered TE in response to ethylene, more than 90% (1788) of the genes are directly (105) or indirectly (1683) regulated by MHZ9. Our study discovers a master regulator MHZ9 for translational control during ethylene response in rice (Fig. 6j).

The N-terminal domain of MHZ9 contains a putative PRP4 domain, which is likely involved in RNA regulation. By using XRNAX-IP-MS, CLIP-seq, RNA-IP, MS2 RNA labeling approach, and RNA-EMSA, we demonstrate that the MHZ9-N-terminal domain binds to the mRNAs of *OsEBF1/2* (Fig. 4). The full-length of MHZ9 is required for OsEIN2-mediated *OsEBF* mRNA association and the P-body localization. It should be noted that, although MHZ9-N could directly bind to the 3′UTR of *OsEBF* mRNA, OsEIN2 is needed for full-length MHZ9-binding to the 3′UTR of *OsEBF* mRNA (Fig. 4f). It is hence likely that, upon interaction with OsEIN2-C through the Q-rich region in MHZ9-C, MHZ9 may alter its conformation to facilitate RNA binding by its N-terminal domain. Meanwhile, the OsEIN2-C/MHZ9/*OsEBF* mRNA complex may be further aggregated in a higher-order manner through the function of the Q-rich region of MHZ9 and phase separation, leading to P-body formation (Fig. 6j). It should be mentioned that, in Arabidopsis, EIN2-C can bind to 3′UTR of *EBF2* mRNA[15], while Li et al.[14] reported that other RNA-binding proteins may be involved to facilitate the association of EIN2-C with 3′UTR of *EBF1/2* mRNA. The present study indicates that OsEIN2 and MHZ9 may form a complex to bind to 3′UTR of *OsEBF1/2*

mRNA for translational control. It is also possible that rice and Arabidopsis may adopt different mechanisms for translational control in ethylene signaling, considering that, several new regulators of ethylene signaling have been identified in rice[18,20,24].

P-body are sites of mRNA turnover and translational repression[44,45]. However, MHZ9 binding to the *OsEBF1/2* mRNA in the P-body may lead to increased mRNA stability and abundance, rather than degradation of those mRNAs (Supplementary Fig. 9a, c). Given that MHZ9 targets hundreds of mRNAs in ethylene (Fig. 4b, Supplementary Data 2–4), the regulation of *OsEBF1/2* mRNA stability by MHZ9 may be indirect or through unknown mechanism. Through polysome profiling assay, we revealed that ethylene provoked a mild repression on the translation of *OsEBF1/2* mRNA, and such repression is dependent on MHZ9 (Fig. 5a, b). Additional studies including the LUC fusion tests, in vitro translation assay, and Western-blot analysis for OsEBF2 and its target OsEIL1 levels in WT and *mhz9*, all demonstrated that MHZ9 targets *OsEBF1/2* mRNA for translational inhibition (Fig. 5, Supplementary Fig. 9). Ribo-seq analysis indicated that, upon 4 h of ethylene treatment, the TE of *OsEBF1* mRNA was significantly inhibited in a MHZ9-dependent manner (Supplementary Data 9). Additional Ribo-seq analysis revealed that, after 8 h of ethylene treatment, the TEs of both *OsEBF1* and *OsEBF2* were reduced by ~52% in WT. In the *mhz9* mutant, such inhibition is substantially disrupted. Based on these studies, we speculate that the *OsEBF1/2* mRNAs aggregated into the P-body may be translationally repressed but not decayed. These features further support the notion that the *EBFs* mRNA may move from polysomes to P-body in ethylene and return to polysomes upon ethylene withdrawal as suggested from the Arabidopsis study[15]. Some P-body mRNAs have been found to reenter to polysomes for translation in animal cells[44,47]. This reversible translational regulation may enable a rapid and energy-saving control of gene expression for adaption. Another possible explanation for the low endogenous *OsEBFs* mRNA abundance in *mhz9* may be owing to the reduced level of OsEIL1 (Fig. 5e, Supplementary Fig. 9d), which may promote *OsEBFs* expression as reported in Arabidopsis[48]. Other possibilities could not be excluded.

The OsEIL1 protein accumulates drastically within two hours of ethylene treatment. MHZ9 is required for this accumulation in both roots and coleoptiles due to the translational repression of *OsEBF1/2*, leading to reduced OsEBF1/2 level and further OsEIL1 accumulation (Fig. 5e and Supplementary Fig. 9d). The function of the preexisting OsEBFs may be quickly blocked through degradation or via other unknown mechanisms. Whether the OsEBFs proteins can be degraded requires further study. It is also possible that the OsEIN2-C and/or OsCTRs may enter the nucleus directly for OsEIL1 accumulation through mechanisms similar to those from Arabidopsis[49,50]. Whether MHZ9 has a role in the nuclear process needs further investigation.

Considering that *Osebf1 Osebf2 mhz9* still exhibits partial ethylene insensitivity in both roots and coleoptiles (Fig. 5g), *OsEBF1/2* mRNAs may not be the only targets for MHZ9. Actually, hundreds of genes can be associated with MHZ9 under ethylene by CLIP-seq and RIP-seq analyses. Taken that MHZ9 is a P-body-localized protein, these MHZ9-binding RNAs may undergo post-transcriptional and/or translational regulation. Investigation of the MHZ9 direct binding genes with altered RNA stability and/or TE, together with the study of indirectly affected genes with altered TE during ethylene response, should disclose more regulatory roles of MHZ9 in ethylene signaling.

Although MHZ9 can bind many mRNAs, how the RNA recognition specificity of MHZ9 is determined remains unclear. It is possible that both the primary nucleotide sequence and the secondary structure of mRNAs could be involved. Further study should reveal relevant features of MHZ9 recognition on its target mRNAs in specific biological processes.

It should be noted that in animal cells, Grb10-interacting GYF protein 2 (GIGYF2) functions as the ribosome quality control (RQC) factor to prime translational repression or co-translational mRNA

decay[51–53], and mRNA binding may be involved[54]. GIGYF2 may recruit CCR4/NOT complex and/or 4EHP for translational repression[54,55]. Other GYF domain proteins also participate in translational control. GYF-1 directs miRNA-mediated translational repression in *Caenorhabditis elegans*[56]. In Arabidopsis, a GYF domain-containing protein Essential for poteX- virus Accumulation (EXA1)/ SMY2-TYPE ILE-GYF DOMAIN- CONTAINING PROTEIN 1 (PSIG1) was found to affect immunity responses through regulation of translation or RNA metabolism[57,58]. Our present study finds the roles of GYF protein MHZ9 in translational control of ethylene responses through direct RNA binding and subsequent translational regulation of *OsEBF1/2* and many other genes. MHZ9 may also affect translation via indirect manners during ethylene responses.

Between the MHZ9 direct binding mRNA targets and MHZ9-mediated TE-significantly-altered genes in ethylene, 105 genes were overlapped, representing a core set of MHZ9-regulated genes in the early ethylene responses (Supplementary Data 10). This core set of genes may further change the translation and/or relevant behaviors of many other mRNAs during ethylene response. It is interesting to note that the negative regulator of gibberellin responses *OsSLR1*[59] resides within this core set of genes, and may be translationally promoted by ethylene (Supplementary Data 10). Another MHZ9-target gene *OsHK3b* (*LOC_Os01g69920*)[60] was remarkably inhibited by ethylene at the translational level, and may encode a putative cytokinin receptor (Supplementary Data 10). From these analyses, a direct cross-talk between ethylene and other hormone signaling may exist at the level of translational regulation through MHZ9 function.

Through investigation of the *mhz9* mutant, multiple changes in phenotypes under field-grown conditions were observed, including dwarf, tiller number decrease, tiller spreading, and grain size reduction (Supplementary Fig. 1). We speculate that MHZ9 may also have roles that are independent of ethylene. MHZ9 actually binds to many other genes in addition to the *OsEBF1/2* mRNAs (Fig. 4b, Supplementary Data 2–4), and many of the MHZ9-bound mRNAs may not be significantly regulated translationally by ethylene (Fig. 6g, h). It should be noted that the *OseEF1α* (*LOC_Os03g08020*) and *OsUBQ5* (*LOC_Os05g06770*) mRNAs were also identified as MHZ9 direct binding mRNA targets (Supplementary Data 10). During the polysome profiling experiments, *MHZ9* mutation apparently reduced the polysome loading of these mRNAs, and such regulation may be ethylene-independent. In our Ribo-seq analysis, the TEs of *OseEF1α* and *OsUBQ5* mRNAs were mildly reduced in the *mhz9* mutant compared with those in WT in an ethylene-independent manner (Supplementary Data 10). If the translation of the *OseEF1α* mRNAs was strongly inhibited in *mhz9* mutants, the global translation and plant growth would be severely affected, which is not the case in *mhz9* plants. This phenomenon indicates that other complementary approaches or unknown mechanisms may be present. Further investigation of the roles of these ethylene-independent genes may reveal more functions of MHZ9 in the regulation of seedling growth and the agronomic traits of field-grown plants.

Together, we discovered a P-body localized translational regulator MHZ9 in rice ethylene signaling. MHZ9 interacts with OsEIN2-C and then binds to *OsEBF1/2* mRNAs for translational inhibition, allowing suppression of OsEBF1/2-mediated proteasomal degradation and activation of downstream transcription factor OsEIL1 for ethylene responses. MHZ9 also exerts a genome-wide translational response. Our study reveals a previously unidentified mechanism, by which MHZ9 modulates ethylene signaling at translational level, and may facilitate improvements of agronomic traits and stress adaptation in rice and other crops.

## Methods
### Plant materials and growth conditions
The rice mutants *mhz9*, *mhz7-1/Osein2-1* were previously identified in our laboratory[3,16]. The T-DNA insertion rice mutant *Osers2* was

purchased from the POSTECH Biotech Center[18]. The rice mutants *Osebf1* (CDS 1014C insertion), *Osebf2* (29-bp deletion at CDS position 435-463), *Osebf1 Osebf2* (CDS 451C insertion for *Osebf1*, and 69-bp deletion at CDS position 827−895 for *Osebf2*), *Osebf1 mhz9* (CDS 452T deletion for *Osebf1*), *Osebf2 mhz9* (10-bp deletion at CDS position 421−430 for *Osebf2*), *Osebf1 Osebf2 mhz9* (CDS 452T deletion for *Osebf1*, and 69-bp deletion at CDS position 827−895 for *Osebf2*) were generated by using CRISPR/Cas9 or crossing. The plants of *Osein2 mhz9*, *OsEIN2-OX/mhz9*, and *OsEIL1-FLAG/mhz9* were generated by crossing. Material propagation and crossing were performed in the Experimental Farm Stations of the Institute of Genetics and Developmental Biology in Beijing from May to October and in Hainan from November to next April. Ethylene treatment was performed as previously described by refs. 16,18. Briefly, after pretreatment at 37 °C, the uniformly germinated rice seeds were placed on stainless steel sieves which were settled in 5.5-L air-tight plastic containers. After growth for 3 days at 28 °C in the dark with treatment of various concentrations of ethylene or 1-Methylcyclopropene (1-MCP), at least 30 seedlings were measured for root and coleoptile length. For short-term treatments, 10 μL/L of ethylene was applied to rice seedlings or tobacco leaves for different time.

### Map-based cloning of *MHZ9*
The mutant *mhz9* was crossed with Indica varieties MH63 and TN1 to generate the F2 mapping populations. The *MHZ9* locus was mapped to chromosome 1 within a 100-kb genomic region. Then the genes in the region were sequenced to identify *MHZ9*. The *MHZ9* mutation site was further confirmed by PCR-based analysis using dCAPS primers (Supplementary Data 12). For genetic complementation assay, genomic sequence of *MHZ9* from the wild-type (2084 bp sequence upstream of ATG codon, 6536-bp genomic coding sequence, and 2000 bp sequence downstream of stop codon) was cloned into the pCAMBIA2300 vector and transformed into *mhz9*.

### Genetic analysis
*Osers2 mhz9* double mutants were generated by crossing *Osers2* mutant in Dongjin (DJ) background and *mhz9* in WT (Nipponbare) background. *MHZ9-OX* was generated by transforming the *MHZ9* coding sequence driven by the native promoter into WT. *MHZ9-OX/Osein2* was generated by crossing *MHZ9-OX* with *Osein2-1*[18]. *OsEIN2-OX/mhz9* was generated by crossing *OsEIN2-OX* (an *OsEIN2* overexpression transgenic line)[16] with *mhz9*. *Osein2 mhz9* double mutants were generated by crossing *Osein2-1* with *mhz9*. *OsEIL1-OX/mhz9* and *OsEIL2-OX/mhz9* were generated by crossing *OsEIL1-OX* (an *OsEIL1* overexpression transgenic line)[27] and *OsEIL2-OX* (an *OsEIL2* overexpression transgenic line)[27] with *mhz9*, respectively.

### Plasmid construction
To generate different MHZ9 truncations for subcellular localization, the coding sequences of *MHZ9* (1–1530 aa), *MHZ9-N* (1–560 aa), *MHZ9-C* (561–1530 aa), *MHZ9-GYF* (502–560 aa), *MHZ9-Q-rich* (866–1234 aa), *MHZ9*-Q-1 (866–1146 aa), *MHZ9*-Q-2 (866–1094 aa), *MHZ9*-Q-3 (866–1020 aa), *MHZ9*-Q-4 (866–975 aa), *MHZ9*△*polyQ* (△896–908 aa), *MHZ9* (△909−943 aa), *MHZ9* (△944−975 aa), *MHZ9* (△976−995 aa), *MHZ9* (△996−1020 aa), *MHZ9* (△909−975 aa), *MHZ9* (△976−1020 aa), *MHZ9* (△896−1020 aa) and *MHZ9* (△896−1258 aa) were cloned into pMDC83 vector. To construct *35S::OsEIN5-mRFP* and *35S::OsEIN2-C-mRFP*, the coding sequence of *OsEIN5* (*LOC_Os03g58060.1*) or *OsEIN2-C* (463−1282 aa) was cloned into pENTR/D-TOPO vector (Invitrogen) and then cloned into pGWB454 (Invitrogen) vector by using Gateway LR Clonase II. To generate *35S::OsDCP2-mRFP*, the coding sequence of *OsDCP2* (*LOC_Os02g56210*) was cloned into pENTR/D-TOPO vector (Invitrogen) and then cloned into pGWB454 (Invitrogen) vector by using Gateway LR Clonase II. For yeast-two-hybrid (Y2H) assays, the coding sequences of *MHZ9-PRP4* (72–233 aa), *MHZ9-GYF*, *MHZ9-C*,

*MHZ9-N* and *MHZ9-GYF-C* (502–1530 aa) were cloned into *EcoR* I/*Nde* I-digested pGADT7 vector. The coding sequence of *OsEIN2-C* (463–1282 aa) was cloned into *EcoR* I/*Nde* I-digested pGBKT7 vector. For BiFC assays, coding sequences of *MHZ9*, *MHZ9-N*, and *MHZ9-C* were cloned into the gateway-compatible BiFC vector BC1-nYFP, and the coding sequence of *OsEIN2-C* was cloned into the gateway-compatible BiFC vector BC2-cYFP by homologous recombination. For domain mapping assays in co-immunoprecipitation (Co-IP) experiments, the coding sequence of MHZ9 was C-terminally tagged with 3×Myc and inserted into *BamH* I-digested pCAMBIA2300-35S vector. To construct *35S::OsEIN2-C-3×FLAG*, the coding sequence of *OsEIN2-C* was C-terminally tagged with *3×FLAG* and inserted into *EcoR* I/*Xba* I-digested pEZS-NL vector. To construct *pMHZ9::MHZ9-GFP*, the putative promoter sequence of MHZ9 (2084-bp sequence upstream of ATG codon) was cloned and N-terminally fused to the coding sequence of *MHZ9*. Then the fusion sequence was C-terminally tagged with GFP and inserted into *BamH* I-digested pCAMBIA2300-35S vector.

For RNA-IP qPCR assays, the coding sequences of *MHZ9-N* and *MHZ9-C* were C-terminally tagged with 3 × Myc and inserted into *BamH* I-digested pCAMBIA2300-35S vector. 5′UTR and 3′UTR sequences of *OsEBF2* mRNA were cloned into gateway-compatible vectors pGWB408 and pGWB409, respectively. The *35S::OsEIN2-GFP* was previously constructed by Ma et al.[18]. For MS2 tethering assays, a *12 × MS2 binding sites* (*12 × MBS*) fragment was amplified from *pSL-12 × MS2* vector and N-terminally fused to 3′UTR of *OsEBF2* mRNA. Then the fusion sequence was inserted into *BamH* I-digested pCAMBIA2300-35S vector. For the expression of GST-MHZ9-N, GST-MHZ9-C, and GST-OsEIN2-C, the coding sequences of MHZ9-N, MHZ9-C, and OsEIN2-C were cloned into *BamH* I/*Sal* I-digested pGEX-6P-1 vector. The biotinylated and non-biotinylated RNA probes were synthesized by Shengyuankemeng (China).

For examination of OsEIL1 protein abundance, the coding sequence of *OsEIL1* was C-terminally tagged with *3 × FLAG* and inserted into *BamH* I-digested pCAMBIA2300-35S vector. For in vitro translation experiments, the coding sequence of *OsEBF2* with or without 3′ UTR of *OsEBF2* was N-terminally tagged with GFP and inserted into *Nco* I-digested pF3A WG (BYDV) Flexi vector (Promega). All the primers used for plasmid construction in Supplementary Data 12.

### Gene expression analysis by real-time PCR
Two-day-old etiolated seedlings were treated with 10 μL/L of ethylene for different time. Total RNAs from shoots or roots were isolated by using TRIZOL reagent (Invitrogen). The cDNAs were synthesized using a Maxima First Strand cDNA Synthesis Kit (Thermo Fisher Scientific) and real-time PCR (RT-qPCR) was conducted by using Thunder Bird SYBR qPCR mix (Toyobo, Japan) with *OsUBQ5* or *OsActin* as an endogenous control for normalization. The primers are listed in Supplementary Data 12. The RT-qPCR analysis was performed with three biological replicates.

### Subcellular localization analysis
For subcellular localization analysis of MHZ9 and its truncations, their coding sequences were fused with *GFP* and transiently expressed in rice protoplasts or in *Nicotiana benthamiana* leaf epidermal cells. The images were taken by using an Axio Imager 2 fluorescence microscope (Carl Zeiss). Excitation/emission wavelengths were set at 488 nm/500–530 nm for GFP and 561 nm/582–654 nm for RFP.

### BiFC assays
For MHZ9 and OsEIN2-C interaction analysis, the BiFC constructs were co-expressed in rice protoplasts of *Osein2 mhz9* double mutant. Yellow fluorescence was detected by using an Axio Imager 2 fluorescence microscope (Carl Zeiss) with 514 nm/525–565 nm excitation/emission wavelengths[18].

### Yeast-two-hybrid assay
Yeast strain Y2H gold cells were co-transformed with the prey and bait constructs. To detect self-activation, the empty prey vector was co-transformed with *pGBKT7-OsEIN2-C*, and the empty bait vector was co-transformed with *AD-MHZ9*, *AD-MHZ9-N*, *AD-MHZ9-C*, *AD-MHZ9-PRP4*, *AD-MHZ9-GYF* or *AD-MHZ9-GYF-C*. By screening on the SD-Trp-Leu medium, the positive transformants were further detected on the SD-Trp-Leu-His medium. The images were taken after incubation for 4 days at 30 °C.

### Immunoblot analysis
For immunoblot analysis, isolated proteins were mixed with SDS-PAGE loading buffer and denatured at 65 °C for 5 min (for detection of OsEIN2-C-3 × FLAG) or in boiling water for 5 min. The protein samples were then separated by SDS-PAGE. Primary antibody dilutions were in PBS containing 3% milk and 0.1% Tween 20. For GFP-fused protein detection, the primary antibody was diluted in Immunoreaction Enhancer Solution I (Toyobo). The primary antibodies used include anti-OsEIN2 (1:10,000)[18], anti-GFP (7G9) (1:5000; M2004, Abmart), anti-histone H3 (1:10,000; nuclear marker; AS10 710, Agrisera), anti-c-Myc (HRP Conjugated, 1:5000; M20019, Abmart), and anti-FLAG (3B9) (1:5000; M20008, Abmart). Secondary goat anti-rabbit or anti-mouse-IgG-horseradish peroxidase (M210011, M210021, Abmart) antibodies were used at 1:20,000 dilutions in PBS containing 3% milk and 0.1% Tween 20. The signals were detected by chemiluminescence method using SuperSignal West Pico kit (34080, Thermo Scientific) or Smart-ECL Super kit (S32500-1, Smart-Lifesciences). When needed, the signal intensities were quantified by ImageJ software with default parameters (National Institutes of Health).

### XRNAX-immunoprecipitation-mass spectrometry (IP-MS)
XRNAX-IP-MS methods were modified from Trendel et al.[41]. The samples was performed by using *35S::MHZ9-N-GFP* transgenic plants treated with 10 μL/L of ethylene and were UV-crosslinked with 600 mJ/cm² at 254 nm for three times, using a UVP CX-2000 Shortwave Ultraviolet Crosslinker (Upland, CA, USA). Add 1 mL of TRIzol per 100 mg powdered tissue sample and lyse the samples by votexing and incubating for 5 min at room temperature (RT). Add 0.2 mL of chloroform per 1 mL of TRIzol reagent used for lysis to induce phase separation. After centrifuging at 7000 × *g* for 10 min at 4 °C, the aqueous phase was removed and the interphase was transferred into a fresh 1.5 mL tube. The interphase was washed with 1 mL of low-SDS buffer [50 mM TRIS (pH 7.4), 1 mM EDTA, 0.1% (w/v) SDS]. Then the interphase was resuspended with 1 mL of low-SDS buffer. After centrifuging at 5000 × *g* for 2 min at RT, the aqueous phase was harvested as elution-1. Repeat washing with the low-SDS buffer and keep the second aqueous phase as elution-2. Then the interphase was washed again with 1 mL of high-SDS buffer [50 mM TRIS (pH 7.4), 1 mM EDTA, 0.5% (w/v) SDS]. After centrifuging at 5000 × *g* for 2 min at RT, the aqueous phase was harvested as elution-3. Repeat washing with the high-SDS buffer and keep the last aqueous phase as elution-4. The four tubes of elution were collected into a fresh 10 mL tube and mixed with equal amount of isopropanol. After centrifuging at 18,000 × *g* for 15 min at 4 °C, the pellet was washed with 1 mL of 70% ethanol twice. The pellet was resuspended with RNase-free water and treated with DNase I to remove DNA contamination. Then the RNA and protein complex was harvested by alcohol precipitation and washed with 1 mL of 70% ethanol twice. The pellet was resuspended in 1 mL of IP buffer [50 mM TRIS (pH 7.4), 150 mM NaCl, 0.5 mM EDTA, 0.5% NP40]. The complex was incubated with 25 μL equilibrated GFP-Trap_A (gta-20, Chromotek) beads for 4 h at 4 °C and washed 4 times with the IP buffer. The beads were treated with 2.5 μL RNase I (Thermo Scientifc, EN0601) and 2.5 μL RNase A (Sigma, R6148) for 24 h. The beads were washed for 4 times with IP buffer. Then the beads were mixed with 25 μL elution

buffer [5 μL of SDS loading buffer (5 ×), 5 μL of DTT (1 M) and 15 μL of RNase-free water] and heated at 70 °C for 15 min. After centrifuging at 2500 × *g* for 5 min at RT, the supernatants were harvested and analyzed with MS by Shanghai Applied Protein Technology Co. Ltd. The RNA-bound peptides in MHZ9 were detected according to the mass shift via mass spectrometry.

## Co-IP assays

Co-immunoprecipitation assays in this study were modified from Ma et al.[18]. Different combinations of constructs were co-expressed in rice protoplasts. After 16 h of incubation in the dark, then total proteins were isolated by IP buffer [50 mM TRIS (pH 7.4), 150 mM NaCl, 0.5 mM EDTA, 0.5% NP40, 50 μM MG132, 2 mM PMSF, 2% (v/v) protease inhibitor cocktail (Sigma)]. The samples were incubated on ice for 15 min with vortexing every 5 minute. The samples were then centrifuged at 13,000 × *g* for 10 min at 4 °C twice. The 10% of the supernatants were saved as input and mixed with equal amount of 2 × SDS-PAGE loading buffer and heated at 65 °C for 5 min. The rest of the supernatants were mixed with 25 μL equilibrated GFP-Trap_A (gta-20, Chromotek) beads or with anti-c-Myc affinity gel (E6654, Sigma-Aldrich). After incubation with rotation for 1 h at 4 °C, the beads were washed 3 times with washing buffer [10 mM Tris-HCl (pH 7.4), 150 mM NaCl, 0.5 mM EDTA, 50 μM MG132, 2 mM PMSF, 1× complete protease inhibitor (Roche)]. Then the beads were harvested by centrifuging at 2500 × *g* for 2 min at 4 °C. The beads were mixed with 40 μL of 2× SDS-PAGE loading buffer and heated at 65 °C for 5 min. The supernatants were harvested and subjected to the further analysis.

## Liquid chromatograph mass spectrometer (LC–MS)

**For identification of MHZ9 interaction protein assays.** The 2-day-old etiolated *35S::GFP-MHZ9* transgenic rice seedlings after treatment with 10 μL/L of ethylene for 8 h were used (*n* = 1). The wild-type (Nipponbare) rice seedlings under the same conditions were used as negative control (*n* = 1). After immunoprecipitation with GFP-Trap_A (gta-20, Chromotek) beads, the immunoprecipitated proteins were eluted by 50 mM glycine (pH 2.5) and neutralized with 1 M Tris buffer (pH 8.0). For XRNAX-IP-MS, the immunoprecipitated proteins were eluted with 25 μL elution buffer [5 μL of SDS loading buffer (5×), 5 μL of DTT (1 M) and 15 μL of RNase-free water] and heated at 70 °C for 15 min.

**In-solution digestion.** In brief, 10 mM of dithothreitol (DTT) was added and followed by incubation at 37 °C for 1.5 h. And 50 mM of iodoacetamine (IAA) was added to alkylate proteins followed by incubation at room temperature in dark for 40 min. Trypsin was then added with the ratio of trypsin to protein at 1:50 (w/w) for digestion at 37 °C for overnight. Trypsin digestion was stopped by the addition of trifluoroacetic acid (1%). The peptides of each sample were desalted on C18 Cartridges (Empore™ SPE Cartridges C18, bed I.D. 7 mm, volume 3 ml, Sigma), concentrated by vacuum centrifugation and reconstituted in 40 μl of 0.1% (v/v) formic acid. The peptide content was estimated by UV light spectral density at 280 nm using an extinctions coefficient of 1.1 of 0.1% (g/l) solution that was calculated on the basis of the frequency of tryptophan and tyrosine in vertebrate proteins[61].

LC–MS/MS Analysis. LC–MS/MS analysis was performed on a Q Exactive mass spectrometer (Thermo Scientific) which was coupled to Easy nLC (Proxeon Biosystems, now Thermo Fisher Scientific). The mass spectrometer was operated in positive ion mode. MS data were acquired using a data-dependent top10 method dynamically choosing the most abundant precursor ions from the survey scan (300–1800 *m/z*) for HCD fragmentation. Automatic gain control (AGC) target was set to 3e⁶, and maximum inject time to 10 ms. Dynamic exclusion duration was 40.0 s. Survey scans were acquired at a resolution of 70,000 at *m/z* 200 and resolution for HCD spectra was set to 17,500 at *m/z* 200, and isolation width was 2 *m/z*. Normalized collision energy was 30 eV and the underfill ratio, which specifies the minimum

percentage of the target value likely to be reached at maximum fill time, was defined as 0.1%. The instrument was run with peptide recognition mode enabled.

**Data analysis.** The MS/MS spectra were searched using Mascot 2.2 (Matrix Science) against the uniprot_Oryza_sativa_168248_20180102. fasta. For protein identification, the following options were used: peptide mass tolerance = 20 ppm; MS/MS tolerance = 0.1 Da; enzyme = trypsin; missed cleavage = 2; fixed modification: carbamidomethyl (C); variable modification: oxidation (M); score cutoff = 20; minimum number of unique peptides for protein identification is 1. For identification of mononucleotide-peptide in XRNAX-IP-MS, the mass shifts were set as 324.18, 347.22, 323.2, 363.22 Dalton for uridine monophosphate (U), adenosine monophosphate (A), cytidine monophosphate (C), guanosine monophosphate (G), respectively.

## RNA-immunoprecipitation (RNA-IP)

RNA-IP methods was modified from Li et al.[14]. For RNA-IP assays with transgenic rice seedlings, seedlings were ground in liquid nitrogen to a fine powder. The powder (~300 mg) was extracted in 1.1 mL extraction buffer [10 mM HEPES (pH 7.4), 100 mM KCl, 2.5 mM MgCl₂, 10% glycerin, 5‰ (v/v) NP40, 1 mM DTT, 100 U/mL RNase inhibitor, 50 μM MG132, 2 mM PMSF, 2% (v/v) protease inhibitor cocktail (Sigma)]. For RNA-IP assays with rice protoplasts, the samples were directly extracted with 1.1 mL extraction buffer. After centrifuging twice at 13,000 × *g* for 10 min at 4 °C, the supernatants were transferred to a fresh 1.5 mL tube as RNA-protein complexes (RNPs). The 100 μL of the supernatant was saved as Input and mixed with 1 mL of Trizol to isolate RNA. The rest was incubated with 25 μL equilibrated GFP-Trap_A (gta-20, Chromotek) beads or with anti- Myc affinity gel (E6654, Sigma-Aldrich). After incubation by rotation for 1–2 h at 4 °C, the beads were washed 10 times with washing buffer [100 mM KCl, 2.5 mM MgCl₂, 10 mM HEPES (pH 7.4), 10% glycerin, 5‰ (v/ v) NP40, 1 mM DTT, 50 μM MG132, 2 mM PMSF, 2% (v/v) protease inhibitor cocktail (Sigma)]. Then the beads were harvested by centrifuging at 2500 × *g* for 2 min at 4 °C. Discard supernatant and add 1 mL of Trizol to the beads to isolate immunoprecipitated RNAs. Input RNAs and IP RNAs were further analyzed by quantitative reverse transcription PCR (RT-qPCR) or sequenced by Huada Gene Company (Shenzhen, China). For RIP-qPCR assays, the immunoprecipitated RNAs were examined by RT-qPCR. The fold enrichment was estimated from its mRNA abundance in the IP product, normalized by its abundance in the total mRNA. And *OsUBQ5* was used for normalization. For RIP-seq assays, the isolated RNA was mixed with the fragmentation buffer and fragmented into short fragments. Then cDNA is synthesized using the RNA fragments as templates with random hexamer. The short fragments (~150 nt) were purified for end reparation and single nucleotide A (adenine) addition. Then the fragments were connected with adapters and selected for PCR amplication. The quantification and qualification of the sample library was performed with Agilent 2100 Bioanaylzer and ABI StepOnePlus Real-Time PCR System.

## Crosslinking and Immunoprecipitation followed by sequencing (CLIP-seq)

CLIP-seq method was modified from Zhao et al.[62]. After UV-crosslink, the powdered sample (~600 mg) was extracted with 2 mL extraction buffer [10 mM HEPES (pH 7.4), 100 mM KCl, 2.5 mM MgCl₂, 10% glycerin, 5 ‰ (v/v) NP40, 1 mM DTT, 100 U/mL RNase inhibitor, 50 μM MG132, 2 mM PMSF, 2% (v/v) protease inhibitor cocktail (Sigma)]. After centrifuging at 13,000 × *g* for 10 min at 4 °C twice, the supernatants were harvested as RNA-protein complexes (RNPs). The RNA-protein complex was immunoprecipitated with 50 μL equilibrated GFP-Trap_M (gtm-20, Chromotek) beads. After incubation by rotation for 2 h at 4 °C, the beads were washed 5 times with washing buffer [100 mM KCl, 2.5 mM MgCl₂, 10 mM HEPES (pH 7.4), 10% glycerin, 5‰ (v/ v) NP40,

50 μM MG132, 2 mM PMSF, 2% (v/v) protease inhibitor cocktail (Sigma)]. Then the beads were treated with 2.5 μL RNase I (Thermo Scientifc™, EN0601) at 37°C for 20 min. The beads were washed 4 times with washing buffer. Then the beads were mixed with 25 μL elution buffer [5 μL of SDS loading buffer (5×), 5 μL of DTT (1 M) and 15 μL of RNase-free water] and heated at 70 °C for 15 min. The supernatants were harvested and the RNA-protein complex was further separated using SDS-PAGE and transferred to nitrocellulose membrane. Membrane pieces corresponding to the RNPs-containing regions were used to recover RNA fragments. The membrane pieces containing RNPs were treated with Proteinase K (26160, Thermo Scientific; 10 μl, 20 mg/ml) at 35 °C for 30 min, and purified with phenol/ chloroform/ isoamyl alcohol (25:24:1). The purified RNA fragments were sequenced by Huada Gene Company (Shenzhen, China) through small-RNA sequencing.

### RNA-EMSA

The potential MHZ9-binding site of *OsEBF2* mRNA was chosen according to our MHZ9 CLIP-seq data. RNA-EMSA was performed with a LightShift chemiluminescent RNA-EMSA Kit (Thermo, USA, 20158) following the manufacturer's instruction. Briefly, purified proteins and unlabeled RNA probe were mixed in 1× REMSA binding buffer [10 mM HEPES (pH 7.4), 20 mM KCl, 1 mM MgCl$_2$, 1 mM dithiothreitol]. After pre-incubation at 25 °C for 20 min, the biotin-labeled RNA probe was added and mixed well. The protein/RNA complexes were incubated at 25 °C for 20 min. Then the protein/RNA complex was separated by the 6% native polyacrylamide gel and transferred to nylon membrane (Thermo, USA) at 4 °C. The protein/ RNA complexes were crosslinked with 120 mJ/cm$^2$ at 254 nm using a UVP CX-2000 Shortwave Ultraviolet Crosslinker (Upland, CA, USA) for 45–60 s exposure using the auto crosslink function. The membrane was blocked in 20 mL of blocking buffer for 15 min at room temperature (RT). Then the blocking buffer was replaced with conjugate/blocking solution and incubated with the membrane for 15 min at RT. The membrane was washed with 1× wash buffer and equilibrated with 15 mL of substrate equilibration buffer for 5 min at RT. Then the membrane was incubated in working buffer and exposed to films. *OsEBF1*-3'UTR probe: CAUUACAUCAUGCUGUUU UUUUUCCAUUGCGUCGUGUUCGGAG. OsEBF2–3'UTR probe: CACC AUGAUUGUUUUUUAGGUUGCCGUAGUGUCCCUUGUCCUUUUUUCU UUACUGC. Twenty-fold or two hundred-fold molar excess of non-biotinylated *OsEBF1* or 2–3'UTR probes were used as competitors.

### In vitro translation assay

In vitro translation assays were performed using the TNT Quick-coupled transcription translation system (Promega Corp.). In brief, the BYDV constructs (2 μg) and the expressed proteins were mixed with the TNT® SP6 High-Yield wheat germ master mix and incubated at 25 °C for 4 h. BYDV construct harboring coding sequence of GFP was used as a control. Enough wheat germ master mix (100 μL) was used to avoid competition between *GFP* and *GFP-OsEBF2*. After incubation, 20 μL of the product was mixed with equal volume of 2× SDS-PAGE loading buffer and heated at 65 °C for 5 min. The product abundance was detected using Western-blot. The remaining products were analyzed through polysome profiling assays to determine the translation efficiency of *OsEBF2* mRNA under different conditions.

### Polysome profiling

Polysome profiling methods were modified from Zhao et al.[62]. In brief, rice seedlings were ground in liquid nitrogen to a fine powder and extracted in 1 mL (per 200 mg of powder) polysomes extraction buffer (PEB) [200 mM Tris-HCl (pH 8.4), 50 mM KCl, 25 mM MgCl$_2$, 1% (w/v) polyoxyethylene (23) lauryl ether (Brij-35), 1% (v/v) Triton X-100, 1% (v/v) octylphenyl-polyethylene glycol (Igepal CA630), 1% (v/v) polyoxyethylene sorbitan monolaurate 20 (Tween 20), 1% (v/v) polyoxyethylene (10) tridecyl ether, 1 mM DTT, 1 mM PMSF, 100 μg/mL

cycloheximide, and 50 μg/mL chloramphenicol]. After centrifuging at 16,000 × $g$ for 15 min at 4 °C for three times, the supernatants were transferred to a fresh 1.5 mL tube. Sample amount was determined by optical density at 260 nm (OD260) using an Eppendorf BioPhotometer 6131 (Hamburg, Germany). Equal amount of sample was transferred to a fresh 1.5 mL tube, and 5% of which was saved and mixed with 1 mL of Trizol to isolate Input-RNA, and the remaining 95% of the sample was separated through a 15–50% sucrose gradient by ultracentrifugation at 213,700 × $g$ (SW41 rotor, Beckman) for 3 h at 4 °C. Then profiling signals were detected using a piston gradient fractionator (Biocomp, B152-002) at 254-nm UV absorbance, which was set to divide the gradient fractions into 15 fractions. The RNA in each fraction was isolated using phenol/ chloroform/isoamyl alcohol (25:24:1). And the relative mRNA abundance of a gene in each fraction was calculated by the ratio between its mRNA abundance in each fraction and that in the input-RNA, and *OsUBQ5* was used for normalization.

### Ribosome footprints analysis

The ribosome-footprint procedure was modified from Zhao et al.[62]. Briefly, 2-day-old etiolated WT and *mhz9* seedlings were treated with or without 10 μL/L of ethylene for 4 h. Total RNA were isolated by using TRIZOL reagent (Invitrogen) for RNA-seq. And the ribosomes were extracted from three grams of rice seedlings as described in polysome profiling. Two biological replicates were prepared for the RNA-seq and Ribo-seq experiment, respectively. The supernatants were further pelleted by ultracentrifugation at 4 °C for 16 h at 192,400 × $g$ (Beckman, 70Ti rotor) through a 30-mL sucrose cushion buffer [200 mM Tris-HCl (pH 8.4), 50 mM KCl, 25 mM MgCl$_2$, 1.75 M sucrose, 1 mM DTT, 100 μg/mL cycloheximide, and 50 μg/mL chloramphenicol]. The pellets were resuspended with RNase I (Thermo Scientifc™, EN0601) digestion buffer [50 mM Tris-HCl (pH 8.4), 100 mM KCl, 20 mM MgCl$_2$, 1 mM DTT, 100 μg/mL cycloheximide, and 50 μg/mL chloramphenicol]. After digestion with RNase I for 2 h at room temperature, ribosome-protected mRNA fragments were separated by ribosome profiling. Ribosome profiling was performed using a 15–50% sucrose gradient and ultracentrifugation at 213,700 × $g$ (SW41 rotor, Beckman) for 3 h at 4 °C. The mono-ribosome (monosome) associated fraction was obtained using a piston gradient fractionator (Biocomp, B152-002) at 254-nm UV absorbance. The monosome protected mRNA fragments were purified using phenol/ chloroform/isoamyl alcohol (25:24:1). The mRNA fragments were further separated through denaturing urea [17% (w/v)] polyacrylamide gel electrophoresis (Urea PAGE), and the -28 nt fragments were recovered for RNA isolation. Then the RNA fragments were applied to small-RNA library construction for Illumina sequencing by Huada Gene Company (Shenzhen, China). The translation efficiency of a gene was calculated as the Ribo-seq reads RPKM (reads per kilobase per million mapped reads)/RNA-seq reads RPKM.

### Bioinformatics analysis

Bioinformatics analysis was performed with the Galaxy public server (https://usegalaxy.eu/).

Raw RNA-seq, RIP-seq, CLIP-seq and filtered Ribo-seq reads (tRNA, rRNA depletion) were first trimmed and quality filtered with cutadapt (https://github.com/marcelm/cutadapt)[63]. Then, all these trimmed and filtered reads were mapped against *Oryza sativa* genome from NCBI (GCF_001433935.1) with TopHat2[64]. The bam files were sorted, indexed and depth calculated with SAMTools[65].

Briefly, the data of RIP-seq and CLIP-seq were loaded into PEA-Kachu with the parameter [windowed approach, DESeq2 method, $p$ value < 0.05, Fold change ≥2]. The overlapped 626 genes between MHZ9-binding genes from CLIP-seq and those from RIP-seq are defined as stringent MHZ9 targets. For peak analysis, the CLIP sites of the 626 target genes were defined as MHZ9-target sites. For motif search, the peak sites were extracted with Samtools, and uploaded to MEME website (https://meme-suite.org/meme/) for further analysis.

For RNA-seq and Ribo-seq analysis, RibORF software was used to identify ORFs and perform quality assessment of Ribo-seq data. Reads mapped to the ORFs were selected for further analysis. The DEseq2, edgeR, and deltaTE[66] was used to identify differentially expressed genes at RNA, translational, and translation efficiency (TE) levels (GCF_001433935.1_IRGSP-1.0_genomic.gff), respectively. The significantly differential genes were selected under the *P* value cutoff 0.05. The statistical corrections for multiple testing (false discovery rate analysis) were also performed and shown in the Supplementary Data 7–10. With the FDR 0.05 cutoff, much higher proportion of ethylene-responsive genes were affected by MHZ9 at both the translational and TE levels (Supplementary Data 8 and 9). Kyoto Encyclopedia of Genes and Genomes (KEGG) analysis was performed using TBtools[67]. Other detailed analysis and data visualization were performed with Python. The primers used in this study are listed in Supplementary Data 12.

## Statistics and reproducibility

No sample size calculation was performed. Sample size of all experiments was decided based on the feasibility of sample collection. The number of replication are indicated in the figure legends. Samples were arranged randomly in related experiments. The blinding was not applied. Because all the experiments were performed without prior knowledge of the final outcome. No data were excluded from our analyses. The results are reliable and reproducible.

## Reporting summary

Further information on research design is available in the Nature Portfolio Reporting Summary linked to this article.

## Data availability

The RNA-IP Seq data generated in this study have been deposited in the NCBI/Sequence Read Archive (SRA) database under accession code PRJNA890558. The CLIP-Seq data generated in this study have been deposited in the NCBI/Sequence Read Archive (SRA) database under accession code PRJNA896502. The Ribosome footprints Seq data generated in this study have been deposited in the NCBI/Sequence Read Archive (SRA) database under accession code PRJNA891450. The mass spectrometry proteomics data generated in this study have been deposited in the ProteomeXchange Consortium via the PRIDE partner repository under accession code PXD041240. All other study data are included in the article and/or supporting information. Source data are provided with this paper.

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

## Acknowledgements

This work is supported by the National Natural Science Foundation of China (31530004, J.-S.Z.; 32000220, Y.Z.; 31600980, C.-C.Y.; 32200257, Y.-H.H.) and the State Key Laboratory of Plant Genomics, IGDB, CAS.

## Author contributions

Y.-H.H., J.-S.Z., C.-C.Y., BM and S.-Y.C. designed the research; Y.-H.H. performed most of the research; J.-Q.H. did the bioinformatic analysis; C.Y. identified the gene and evaluated the yield and agronomic traits of mhz9 plants; X.-K.L. edited Fig. 6 a–c; B.M. and Q.X. isolated the mutant; all authors including Y.-H.H., J.-S.Z., C.-C.Y., B.M., S.-Y.C., J.-Q.H., X.-K.L., B.M., Q.X., W.-Q.C., H.Z., R.Z., X.Z., Y.Z., W.W., J.-J.T., W.-K.Z. and W.-F.Q. contributed to material preparation, data analysis and discussion; Y.-H.H. and J.-S.Z. wrote the article.

## Competing interests

The authors declare no competing interests.
