## [Peer Review File · Nature Communications]

A translational regulator MHZ9 modulates ethylene signaling in riceREVIEWER COMMENTS

Reviewer #1 (Remarks to the Author):

Dr Zhang et al., identified a glycine-tyrosine-phenylalanine (GYF) domain-containing protein MHZ9, which interacts with OsEIN2-C in P-body and directly binds to the OsEBF1/2 mRNAs for translational repression, leading to an ethylene insensitivity in rice. It is an interesting research that identified a new component for translational regulation in the ethylene response that mediated by EIN2, leading to a deep understanding of translational regulation in ethylene signaling.

There are some main concerns:

1. The colocalization results from Fig3b, and 3c are not from the same cells. The result in Fig 3b is from tobacco leaves, while the result in fig 3C is from Arabidopsis protoplast. The best comparison is from the same testing system. In addition, the co-localization of three proteins is required.
2. Fig 3i, the IP product of Q-3-GFP is significantly smaller than the input Q-3-GFP in protein size. More accurate and careful experiment should be conducted for the right protein.
3. Fig 4h, a dose dependent binding activity is necessary to provide more solid evidence that MHZ9-N binds to the 3' UTR of EBF1/2 mRNA.
4. fig 5c, EIN2 is required for the translation, it is hard to understand that the expression of EBF2 proteins are lower in ein2 mutant than that in mhz9 mutant both with or without the presence of 3' UTR. Suppose EIN2 represses the EBF1/2 translation, leading to accumulation of EIN3, therefore, in ein2 mutant, EBF2 protein should be accumulated to about or even a higher than in mhz9 mutant.

Reviewer #2 (Remarks to the Author):

This article presents many thorough results that document how a new gene product, the GYF-domain protein MHZ9, contributes to ethylene signaling in rice. Overall, this is a very thorough paper with an impressive array of different assays documenting how MHZ9 is important alongside EIN2 for ethylene signaling in etiolated seedlings. Overall the cloning of the new component of ethylene signaling and its characterization as a posttranscriptional regulator of EBF1/2 are important and move the field forward.

I will summarize what are the key findings in the paper. The conclusions generally flow easily from the data. However, on the translational control experiments, I will present observations that show that the interpretations by the authors are not as straightforward as they may seem and suggest that the data be reanalyzed. I also have a fundamental concern with the model presented at the end, which should be discussed as a basis for future work.

In rice and Arabidopsis, it is known that ethylene regulates translation. It does so via EIN2. In Arabidopsis the EIN2 CTD represses the translation of the F-box proteins EBF1/2, which are the FBPs for EIN3. Thus, activation of EIN2 also activates EIN3. Here authors show in rice that EIN2 does not act alone but through an RNA binding GYF domain protein, MHZ9, which was newly cloned in this work. MHZ9 was identified by a large screen for ethylene-insensitive mutants in rice. MHZ9 interacts with the 3'UTR of the rice EBF1 and EBF2 mRNAs, which in rice are thought to target the EIL1 transcription factor for degradation, rather than EIN3 as in Arabidopsis. MHZ9 is found in P-bodies which are cytosolic sites for translational repression, mRNA decapping and mRNA turnover.

The work then addresses the hypothesis that MHZ9 inhibits EBF1/2 mRNA translation, which would cause EIL1 to accumulate. The work presents several assays to test whether MHZ9 represses EBF1/2 translation, which I will comment on later. They also present evidence using ribosome footprinting that MHZ9 is the link that allows ethylene to regulate translation of numerous mRNAs.

I will summarize the early results, which I found thorough and convincing.

Authors cloned MHZ9, and found that the loss of function mutant has a 1bp deletion causing a premature termination codon. The mutant is strongly ethylene-resistant, measuring root and coleoptile growth. They did RNAi and complementation to confirm they cloned the correct gene (Fig. S1-S3).

A key result was that MHZ9-overexpression caused ethylene-agonist phenotypes, which were abrogated by 1-

MCP, an inhibitor of ethylene perception. Hence the action of MHZ9 is indeed ethylene dependent (Fig 1d). These are clear results, especially in the root growth assay, where ethylene requires MHZ9 to block growth.

They then studied genetic interactions of MHZ9 and ERS2 (ethylene receptor), EIN2 and EIL1, concluding that MHZ9 functions between EIN2 and EIL1 in the ethylene signaling pathway (Fig. 2). These results also look clear to me. MHZ9 and EIN2 interact with each other by a variety of assays, and they localize independently to P-bodies (Fig. 3).

The N-terminal domain of MHZ9 binds many mRNAs, but especially EBF1/2, and the binding is EIN2-dependent. This was shown with many assays such as RIP-Seq, CLIP-Seq and colocalization with MS2-GFP. They also have an assay that directly identifies which bases are bound by which peptide in MHZ9, called XRNAX-IP-MS; the description of this method seems a bit opaque, please check if it is complete.

These results support the claims made earlier. At this point, because P-bodies are sites of mRNA decapping and turnover, I expected that they examine mRNA levels and mRNA stability for endogenous EBF1/2 mRNAs in air (hypothesis: high and stable) versus in ethylene (hypothesis: lower and unstable). The mRNA levels actually behave the opposite from what is expected (ethylene boosts mRNA levels of EBF1 slightly and does not affect EBF2), but stability results were not presented. I would like the authors to carefully discuss these important observations. If they trust their results, then they should draw the logical conclusions from them (see comments on the model below).

They then examined whether MHZ9 affects the translational efficiency of EBF1/2 mRNA, again using an impressive variety of assays. I commend the authors for this extremely thorough approach, even though, as is common with translation assays, the results merit scrutiny.

1. On the polysome profiling data (Fig. 5a) I can see the trend that the authors are focusing on, but they need to properly interpret several other obvious signals in this experiment.

a. It is most striking that a very small proportion of the total EBF1/2 mRNAs is recovered in the polysomal fractions, generally well below 10%. I would like to know whether other generic mRNAs, for example eEF1alpha, are recovered equally poorly or not. This would help to confirm whether indeed EBF1/2 mRNAs are very poorly translated (or especially unstable under these experimental conditions).

b. A technical concern - it is not clear how EBF1/2 mRNA was quantified because the legend says data are expressed relative to the abundance of UBQ5, rather than as % total. It is important to note that the endogenous UBQ5 mRNA must be distributed in its own, non-uniform, way across the gradient! Thus, any subjective peak in EBF1/2 mRNA in the gradient may simply be due to the fact that UBQ5 is LOW in that fraction. This should be more transparent, by simply showing the results of UBQ5 and EBF1/2 as raw CT values against total RNA. The conclusions should hold up against showing the data in this way.

c. Third, in the WT the EBF1/2 mRNAs are detected primarily in only one or two fractions. And these are fractions with very high polysome loading, i.e. ~9+ ribosomes. This is generally an uncommon result because typical mRNAs are spread across the entire gradient. Authors should consider this, and that these EBF1/2 ribosomes may be actively translating or may be stalled.

d. Fourth, as stated by the authors, ethylene seems to inhibit polysome loading of both EBF1 and EBF2. I see what the authors are seeing, except for my caveats above; this result would be stronger if ~eEF1alpha control mRNA was shown.

e. Finally, the most striking result in this experiment is that the mhz9 mutants have a lot of EBF1/2 mRNA in large polysomes and at the bottom of the gradient! This is a fairly unusual result, and again we need to see UBQ5 (it may be almost absent in fraction 15) and another reference mRNA such as eEF1alpha (maybe it behaves the same as EBF1/2?). I also wished the authors had not ignored fractions 1-7 of their gradients. This would help to distinguish whether MHZ9 represses polysome loading or destabilizes the mRNAs.

2. The subsequent luciferase reporter assays are technically less questionable, although the results are also complicated. I agree that MHZ9 and especially EIN2 seem to work together to suppress the translation of EBF2 and also EBF1 through its 3'UTR (Fig 5b), but MHZ9 has a small effect in this assay. Also, the authors should not ignore that MHZ9 and EIN2 also boost mRNA levels of the LUC-3'UTR reporter mRNA and of endo-EBF1/2 mRNAs (Fig. S6b). This fairly conclusive result could be described more transparently (lines 244 ff) and the implication for the model at the end should be discussed. If MHZ9 boosts RNA level but represses translation, it hard to see how this would cause EBF1/2 levels to drop, as envisaged by the model at the end. Perhaps the

EBF1/2 mRNA in the P-bodies is stabilized but remains untranslated.

3. Ribo-Seq and RNA-Seq experiments. The ribo-seq experiment appears to have yielded some data consistent with a global role of MHZ9 in translational control. However, the underlying raw data are peculiar.

a. In the first panel, Fig. 6a, it appears that 4 hours of ethylene causes only very subtle changes in mRNA levels; almost no mRNAs changed by more than 2-fold! This is very peculiar because there are plenty of genes known to respond robustly to ethylene, including in Fig. 1E/F of this paper *The Plant Cell*, Volume 27, Issue 4, April 2015, Pages 1061-1081, <https://doi.org/10.1105/tpc.15.00080> by the same authors.

b. Second, in ribo-seq experiments it is common that ribosome footprint counts correlate closely with mRNA levels. In this experiment the correlation is extremely low. It appears that nearly every mRNA that is repressed at the transcript level is repressed less (or even induced) at the footprint level, indicating translational enhancement. Together with the previous point, this again suggests that the transcript-level response to ethylene was undercounted.

c. Third, the numerical values listed in Line 314 have no relation to the corresponding figure 6a. This is difficult to understand.

d. Fourth, line 317 counts genes with significant changes under ethylene, but the methods are inappropriate. Line 768: The methods say nothing about statistical corrections for multiple testing (false discovery rate analysis). In fact, checking for statistical significance by T-test (line 788) is not appropriate and should not be done. Any claims for statistical significance by t-test are invalid.

e. Fifth, the data in Fig. S7b look as if they come from a single gene, but it is not listed which. The legend suggests that they are aggregated across all genes. However, the erratic pattern of footprints suggest that this is not the case.

Taken together, although the ribo-seq figures suggest the very straightforward conclusion that MHZ9 is responsible for the global translational response to ethylene, the underlying data raised a lot of questions that remain to be answered in order to have faith in the result.

Fig. 6i. The model. What this model does not show is what happens to the EBF1/2 protein in ethylene. The model suggests that all the EBF1/2 protein present in 'air' disappears extremely rapidly when the cell is exposed to ethylene. How this happens is not at all clear. Most proteins have a half life of at least several hours. During this time translation of new EBF1 protein may indeed be stalled by the translational control mechanism, but the preexisting EBF1 protein would presumably still linger around for many hours and continue to degrade EIL1. But this does not seem to be happening because EIL1 accumulates very fast within 2h of ethylene in an MHZ9 dependent manner (Fig. S6c) while EIL1 mRNA barely changes (Fig. S6d). The model therefore predicts that EBF1/2 is an extremely unstable protein. Has this been shown to be the case? Or is this a hypothesis to be addressed in future work?

Minor points:

Did your ribo-seq experiment confirm the prediction from Fig. 5a, that EBF1 has very low translational efficiency?

Fig S1: In panel g the II did not get printed properly.

Fig. S3b: In panel b it is peculiar that IAA20 and ERF2 are not induced by ethylene in the wild type, when in Fig. 1b they were. How to explain this?

Line 720: 'microgram' is probably incorrect.

Line 748: '30mL' is probably incorrect.

Line 776: 'datas'; this word does not exist yet. Why? Because 'Data' is already a plural. Of course, after years of grammatically abusing 'data' as a singular noun, it is only logical that at some point in time 'datas' would appear on the scene.

Discussion

The discussion section introduces many additional results, which I feel could be held back for another paper once the current paper is sound and published.

Writing:

While the first part of the results was edited well and easily readable, the later part is written in a more convoluted manner. "The easier it is to read the harder it is to write".

Example for how to edit the writing: "In WT, an obvious reduction of TEs of OsEBF1/2 mRNAs were detected, especially at later fractions after ethylene treatment for 2 h compared to those in air."  Better: In the wild type, ethylene reduced the abundance of EBF1/2 mRNAs in fraction 14, which harbors large polysomes. There are many passages that deserve editing.

Reviewer #3 (Remarks to the Author):

This manuscript reports the interesting finding of a novel translational regulator called MHZ9, which plays a significant role in ethylene hormone signaling in rice. Ethylene signaling is important in many agronomically important traits in plants, including responses to biotic and abiotic stresses. Significantly, the results in this manuscript connect a key protein in ethylene signaling (EIN2) with the control of downstream gene expression in ethylene signaling. Moreover, the mechanism involving MHZ9 appears to be novel and may have broader implications for how translation of mRNAs could be regulated.

In general, the experiments appear to be technically sound, but I feel the reporting of the experiments and data could be made clearer as described below. Overall, the manuscript provides a nice story starting with a mutant and leading to gene identification and elucidation of protein function with mechanistic insights.

Suggestions and questions:

As stated in the Abstract, "regulation of ~ 90% of the translationally affected genes by ethylene requires MHZ9 function (lines 33-34)" and in the Discussion, "MHZ9 directly regulates the translation efficiencies (TEs) of its binding targets, and regulation of more than 90% of the genes with ethylene-altered TE levels is dependent on MHZ9 (lines 365-368)" I think the authors could make clearer statements that they believe such regulation is direct, i.e., does not occur via MHZ9's control of OsEIL1/2 transcription factor protein levels (which was what I thought upon initially reading the manuscript).

On the other hand, it would be helpful if the authors could spell out in the Results and Discussion precisely why they have ruled out the possibility of enhanced transcription/translation due to the accumulation of the OsEIL1/2 transcription factors that activate downstream ethylene-response genes.

The epistasis results for MHZ9 and OsEBF1/2 shown in Figure 5f are unexpected. This unexplained finding should be discussed in the Discussion. For example, could there be a role for MHZ9 in the nucleus where EIN2-C is also found? Related to this, on Line 292, I suggest changing "largely insensitive" to "partially insensitive".

Line 133-134: The following sentence needs a bit of clarification: "All the results indicate that MHZ9 and OsEIN2 are likely mutually dependent in ethylene signaling." What is the specific meaning of "mutually dependent", and what is the rationale for this conclusion? For example, I would expect that ethylene insensitivity from a dominant mutant ethylene receptor would show the same complete insensitivity when paired with the mhz9 mutation. Would this indicate that the ethylene receptor and MHZ9 are "mutually dependent"?

For the section on "Genetic interaction of MHZ9 with ethylene signaling components in rice", was there a technical reason why genetic interactions between MHZ9 and OsCTR1 were not examined? Interaction between MHZ9 and OsCTR1 would make good sense given that the corresponding loss-of-function mutants should have opposite phenotypes.

Along the same lines, the authors could add "either upstream or downstream of OsEIL1/2" at the end of the following sentence (lines 138-139), "From all the genetic analyses, MHZ9 may function downstream of OsERS2, and be required for OsEIN2 and OsEIL1/2 function in ethylene signaling". This would help to clarify that the exact placement of MHZ9 in the pathway is not fully established by the genetic analyses.

The introduction describes the ethylene signaling pathway as known in Arabidopsis. Since this manuscript focuses on rice, it would be relevant to know the similarities and differences that have been identified in rice compared to the ethylene signaling pathway of Arabidopsis.

The gene names of all ethylene signaling components are fully spelled out in the Introduction, with the exception of the subject of this manuscript, "mhz9". It would be helpful for the reader to know what the name "mhz" stands

for.

Is there an MHZ9 homolog in Arabidopsis? The answer is not explicitly provided but is relevant to the Discussion on lines 381-388.

I understand that the manuscript is likely streamlined to fit within the word limit for the journal, but in some cases, pertinent information appears to be absent. For example, it is unclear how many times experiments were repeated.

Line 209: I'm not familiar with the "MS2 tethering assay". If there is space, perhaps a brief explanation would be helpful in the Results or Methods. I did not see a direct description of the MS2 tethering assay under the Methods section.

Line 245, the "optimized dual luciferase assay" should be explained. What is a dual luciferase assay, and in what way has it been optimized?

For the dual luciferase assay, it is unclear to just say "in mhz9 and Osein2" (line 251 plus many other sentences). The use of protoplasts for these experiments needs to be specified, e.g., "in mhz9 and Osein2 protoplasts". Related to this, throughout the manuscript, the authors refer to just "mhz9", when it would greatly help the reader to see a noun after "mhz9", such as "mhz9 protoplasts", "mhz9 gene", "mhz9 mutant", or "mhz9 protein" or "mhz9 plants". For example, on line 98, please change "The mhz9 was isolated" to "The mhz9 mutant was isolated."

In Figure 5e, the input amount of protein is not known.

It looks like complete statistical analyses are missing for the data in Figure 2b.

The authors should cite a reference for this portion of line 145: "whose Arabidopsis homologue EIN5 is a P-body marker", as they did for OsDCP2 on line 149.

In Figure 3f, what do the "x1", "x10", and "x10²" refer to? If they indicate dilutions of yeast cells, then the exponent should be a negative number, correct? In the legend, I suggest spelling out "Y2H".

In Figure 3i, white asterisks would be easier to see than the red asterisks.

Line 189: Spell out "long intergenic non-coding RNAs" when first using the term "lincRNAs"

On line 287, it would be helpful to specify the type of mutants used in the epistasis analysis, such as "single loss-of-function mutants"

The grammar in the manuscript needs some improvement, especially with respect to incorrectly leaving out the word "the" in many places or leaving out "a" or "an".

Lines 118 and 122, change to: "field conditions".

Typos:

Line 420: "MHZ9-regualted"

Line 93, In "Our study identifies a master translational regulator of MHZ9", I believe the word "of" should be deleted, given that MHZ9 itself is the master translational regulator.

Response to the reviewers' comments

For Reviewer #1:

Dr Zhang et al., identified a glycine-tyrosine-phenylalanine (GYF) domain-containing protein MHZ9, which interacts with OsEIN2-C in P-body and directly binds to the OsEBF1/2 mRNAs for translational repression, leading to an ethylene insensitivity in rice. It is an interesting research that identified a new component for translational regulation in the ethylene response that mediated by EIN2, leading to a deep understanding of translational regulation in ethylene signaling.

【Response】 Thank you for the comments.

There are some main concerns:

1, The colocalization results from Fig3b, and 3c are not from the same cells. The result in Fig 3b is from tobacco leaves, while the result in fig 3C is from Arabidopsis protoplast. The best comparison is from the same testing system. In addition, the co-localization of three proteins is required.

【Response】 Thanks for your comments and advice. According to the suggestions, we have performed additional experiments to support the co-localization of MHZ9 with OsDCP2, and the co-localization of OsEIN5, OsEIN2-C, and MHZ9 in tobacco leaf assay. The added results have been integrated into our revised manuscript (Line 172-175), and Fig. 3b.

2, Fig 3i, the IP product of Q-3-GFP is significantly smaller than the input Q-3-GFP in protein size. More accurate and careful experiment should be conducted for the right protein.

【Response】 Thank you for the comments. We have corrected it in the revised Fig. 3h. The band corresponding to each of the truncated proteins is marked by a triangle. The upper bands with higher molecular weights than expected may arise from some protein modifications.

3, Fig 4h, a dose dependent binding activity is necessary to provide more solid evidence that MHZ9-N binds to the 3' UTR of EBF1/2 mRNA.

【Response】 Thanks for your advice. According to the suggestion, we have performed additional experiments to further prove the binding of MHZ9-N to the 3' UTR of *OsEBF1/2* mRNA. The results have been integrated into our revised figures (supplementary Fig. 9) and the text (Line 252-253).

4, fig 5c, EIN2 is required for the translation, it is hard to understand that the expression of EBF2 proteins are lower in *ein2* mutant than that in *mhz9* mutant both with or without

the presence of 3' UTR. Suppose EIN2 represses the EBF1/2 translation, leading to accumulation of EIN3, therefore, in ein2 mutant, EBF2 protein should be accumulated to about or even a higher than in mhz9 mutant.

【Response】 Thank you for the comments. Based on our findings, OsEIN2 and MHZ9 may form a complex to bind to 3'UTR of *OsEBF1/2* mRNA for translational repression in ethylene signaling (Fig. 4 and Fig. 5). Since MHZ9 directly binds to *OsEBF1/2* mRNA (Fig. 4h and Supplementary Fig. 9), it is possible that other OsEIN2-independent pathways may repress *OsEBF1/2* mRNA expression through MHZ9, and the MHZ9 binding sites in the coding region of *OsEBF1/2* mRNA (Fig. 4c) may also contribute to this regulation. We have made revisions in the text (Line 310-314) about this result.

Thank you very much for all your comments and suggestions.

For Reviewer #2:

This article presents many thorough results that document how a new gene product, the GYF-domain protein MHZ9, contributes to ethylene signaling in rice. Overall, this is a very thorough paper with an impressive array of different assays documenting how MHZ9 is important alongside EIN2 for ethylene signaling in etiolated seedlings. Overall the cloning of the new component of ethylene signaling and its characterization as a posttranscriptional regulator of EBF1/2 are important and move the field forward.

【Response】 Thank you for the comments.

I will summarize what are the key findings in the paper. The conclusions generally flow easily from the data. However, on the translational control experiments, I will present observations that show that the interpretations by the authors are not as straightforward as they may seem and suggest that the data be reanalyzed. I also have a fundamental concern with the model presented at the end, which should be discussed as a basis for future work.

【Response】 Thank you for the comments. We have performed additional experiments and reanalyzed our data regarding the translational control experiments. The results support that MHZ9 could regulate ethylene signaling at the translational level. We also refined the model (Fig. 6j) according to your suggestions and the discussion has also been revised accordingly (Line 413-419, 439-462).

In rice and Arabidopsis, it is known that ethylene regulates translation. It does so via EIN2. In Arabidopsis the EIN2 CTD represses the translation of the F-box proteins EBF1/2, which are the FBPs for EIN3. Thus, activation of EIN2 also activates EIN3. Here authors show in rice that EIN2 does not act alone but through an RNA binding

GYF domain protein, MHZ9, which was newly cloned in this work. MHZ9 was identified by a large screen for ethylene-insensitive mutants in rice. MHZ9 interacts with the 3'UTR of the rice EBF1 and EBF2 mRNAs, which in rice are thought to target the EIL1 transcription factor for degradation, rather than EIN3 as in Arabidopsis. MHZ9 is found in P-bodies which are cytosolic sites for translational repression, mRNA decapping and mRNA turnover.

The work then addresses the hypothesis that MHZ9 inhibits EBF1/2 mRNA translation, which would cause EIL1 to accumulate. The work presents several assays to test whether MHZ9 represses EBF1/2 translation, which I will comment on later. They also present evidence using ribosome footprinting that MHZ9 is the link that allows ethylene to regulate translation of numerous mRNAs.

I will summarize the early results, which I found thorough and convincing.

Authors cloned MHZ9, and found that the loss of function mutant has a 1bp deletion causing a premature termination codon. The mutant is strongly ethylene-resistant, measuring root and coleoptile growth. They did RNAi and complementation to confirm they cloned the correct gene (Fig. S1-S3).

A key result was that MHZ9-overexpression caused ethylene-agonist phenotypes, which were abrogated by 1-MCP, an inhibitor of ethylene perception. Hence the action of MHZ9 is indeed ethylene dependent (Fig 1d). These are clear results, especially in the root growth assay, where ethylene requires MHZ9 to block growth.

They then studied genetic interactions of MHZ9 and ERS2 (ethylene receptor), EIN2 and EIL1, concluding that MHZ9 functions between EIN2 and EIL1 in the ethylene signaling pathway (Fig. 2). These results also look clear to me. MHZ9 and EIN2 interact with each other by a variety of assays, and they localize independently to P-bodies (Fig. 3).

The N-terminal domain of MHZ9 binds many mRNAs, but especially EBF1/2, and the binding is EIN2-dependent. This was shown with many assays such as RIP-Seq, CLIP-Seq and colocalization with MS2-GFP. They also have an assay that directly identifies which bases are bound by which peptide in MHZ9, called XRNAX-IP-MS; the description of this method seems a bit opaque, please check if it is complete.

【Response】 Thank you for the comments. Detailed description of the XRNAX method has been integrated into the revised method section (Line 653-685).

These results support the claims made earlier. At this point, because P-bodies are sites of mRNA decapping and turnover, I expected that they examine mRNA levels and mRNA stability for endogenous EBF1/2 mRNAs in air (hypothesis: high and stable) versus in ethylene (hypothesis: lower and unstable). The mRNA levels actually behave

the opposite from what is expected (ethylene boosts mRNA levels of EBF1 slightly and does not affect EBF2), but stability results were not presented. I would like the authors to carefully discuss these important observations. If they trust their results, then they should draw the logical conclusions from them (see comments on the model below).

【Response】 Thank you for the comments. We have performed additional experiments to detect whether MHZ9 affects the RNA stability (using transcription inhibitor cordycepin) of endogenous *OsEBF1/2* mRNAs. Based on our results, *MHZ9* mutation significantly reduced the *OsEBF1/2* mRNA stability and abundance (Supplementary Fig. 10a, c), suggesting that the *MHZ9* may facilitate the *OsEBF1/2* mRNA stability and abundance. These results have been integrated into the result section (Line 265-271) and discussed in the discussion section (Line 439-453).

They then examined whether MHZ9 affects the translational efficiency of EBF1/2 mRNA, again using an impressive variety of assays. I commend the authors for this extremely thorough approach, even though, as is common with translation assays, the results merit scrutiny.

1. On the polysome profiling data (Fig. 5a) I can see the trend that the authors are focusing on, but they need to properly interpret several other obvious signals in this experiment.

a. It is most striking that a very small proportion of the total EBF1/2 mRNAs is recovered in the polysomal fractions, generally well below 10%. I would like to know whether other generic mRNAs, for example eEF1alpha, are recovered equally poorly or not. This would help to confirm whether indeed EBF1/2 mRNAs are very poorly translated (or especially unstable under these experimental conditions).

【Response】 Thank you for the comments. The appeared poor recovery of *OsEBF1/2* mRNAs in the polysomal fractions is due to that these values are calculated by using the *OsUBQ5* as a reference gene. We repeated the experiments and analyzed the data by using the raw CT values of this mRNA against its total RNA, and ~30% of the total *OsEBFs* mRNA were recovered in the polysomal fractions (10-16). Similar percentages of the polysome-associated mRNAs were recovered for the reference genes. It should be noted that the translation of *OseEF1α* in *mhz9* seems to be inhibited by MHZ9 mutation or by unknown mechanisms. The other two reference mRNAs, *OsUBQ5* and *OsGAPDH*, are less affected by MHZ9 (Fig. 5b). These results have been incorporated into the Fig. 5b and the text at Line 273-281.

b. A technical concern - it is not clear how EBF1/2 mRNA was quantified because the legend says data are expressed relative to the abundance of UBQ5, rather than as % total. It is important to note that the endogenous UBQ5 mRNA must be distributed in its own, non-uniform, way across the gradient! Thus, any subjective peak in EBF1/2 mRNA in the gradient may simply be due to the fact that UBQ5 is LOW in that fraction.

This should be more transparent, by simply showing the results of UBQ5 and EBF1/2 as raw CT values against total RNA. The conclusions should hold up against showing the data in this way.

【Response】 Thank you very much for your advice. Based on your suggestions, we repeated the polysome profiling assay, and all the fractions were collected. The translated mRNAs were detected by using the raw CT values of this mRNA against its total RNA. And the distribution of each mRNA for each genes (*OsEBF1/2*, and three references *OseEF1 α* , *OsUBQ5*, and *OsGAPDH*) was shown for comparison in the revised Fig. 5b. Other parts were also revised accordingly.

c. Third, in the WT the EBF1/2 mRNAs are detected primarily in only one or two fractions. And these are fractions with very high polysome loading, i.e. ~9+ ribosomes. This is generally an uncommon result because typical mRNAs are spread across the entire gradient. Authors should consider this, and that these EBF1/2 ribosomes may be actively translating or may be stalled.

【Response】 Thank you for your comments. We have repeated the experiments and analyzed the data according to your suggestions. The *OsEBF1/2* mRNAs were distributed across the entire gradient. The reference mRNAs were also shown in separate panels for comparison and clarity. The results have been integrated into our revised figures (Fig. 5a-b) and the text (Line 273-281).

d. Fourth, as stated by the authors, ethylene seems to inhibit polysome loading of both EBF1 and EBF2. I see what the authors are seeing, except for my caveats above; this result would be stronger if ~eEF1 α control mRNA was shown.

【Response】 Thank you for your comments. We have performed additional experiments, and the data were analyzed based on your suggestions. The profiles of the *OsEBF1/2* and the *OseEF1 α* (and the other two reference genes) are shown separately in different panels for comparison. The results can be found in the revised figures (Fig. 5a-b) and manuscript (Line 273-281).

e. Finally, the most striking result in this experiment is that the *mhz9* mutants have a lot of EBF1/2 mRNA in large polysomes and at the bottom of the gradient! This is a fairly unusual result, and again we need to see UBQ5 (it may be almost absent in fraction 15) and another reference mRNA such as eEF1 α (maybe it behaves the same as EBF1/2?). I also wished the authors had not ignored fractions 1-7 of their gradients. This would help to distinguish whether MHZ9 represses polysome loading or destabilizes the mRNAs.

【Response】 Thanks for your advice. We have repeated the polysome profiling assay, and all the fractions were collected for further analysis. The profiles of the *OsEBF1/2* and the *OseEF1 α* , *OsUBQ5* and *OsGAPDH* are shown separately in different panels

for comparison. The results can be found in the revised figures (Fig. 5a-b) and manuscript (Line 273-281).

2. The subsequent luciferase reporter assays are technically less questionable, although the results are also complicated. I agree that MHZ9 and especially EIN2 seem to work together to suppress the translation of EBF2 and also EBF1 through its 3'UTR (Fig 5b), but MHZ9 has a small effect in this assay. Also, the authors should not ignore that MHZ9 and EIN2 also boost mRNA levels of the LUC-3'UTR reporter mRNA and of endo-EBF1/2 mRNAs (Fig. S6b). This fairly conclusive result could be described more transparently (lines 244 ff) and the implication for the model at the end should be discussed. If MHZ9 boosts RNA level but represses translation, it hard to see how this would cause EBF1/2 levels to drop, as envisaged by the model at the end. Perhaps the EBF1/2 mRNA in the P-bodies is stabilized but remains untranslated.

【Response】 Thanks for your comments. Additional experiments have been performed to elucidate the role of MHZ9 in regulation of *OsEBF1/2* mRNA. MHZ9 binding to the *OsEBF1/2* mRNA in P-body is more likely to increase the mRNA stability and abundance, rather than degradation of those mRNAs as judged from the transcription inhibitor (cordycepin) tests (Supplementary Fig. 10a and 10c). It is possible that MHZ9 may stabilize the *OsEBF1/2* mRNAs in P-body and inhibit their translation, and reenter into the polysomes upon removal of ethylene. These have been incorporated into the discussion part at Line 439-453.

3. Ribo-Seq and RNA-Seq experiments. The ribo-seq experiment appears to have yielded some data consistent with a global role of MHZ9 in translational control. However, the underlying raw data are peculiar.

a. In the first panel, Fig. 6a, it appears that 4 hours of ethylene causes only very subtle changes in mRNA levels; almost no mRNAs changed by more than 2-fold! This is very peculiar because there are plenty of genes known to respond robustly to ethylene, including in Fig. 1E/F of this paper *The Plant Cell*, Volume 27, Issue 4, April 2015, Pages 1061-1081, <https://doi.org/10.1105/tpc.15.00080> by the same authors.

【Response】 Thank you for the comments. We reanalyzed our RNA-seq data using a different software DEseq2 with a threshold of genes (both RNA-seq and Ribo-seq reads > 0 RPM vs > 10 RPM in our previous analysis). In this way, 2717 genes are significantly ($P < 0.05$) altered by ethylene. Among these, there are 162 ($P < 0.05$) and 149 (FDR < 0.05) genes with more than 2-fold changes in transcript abundance. The results have been integrated into the revised Fig. 6a and Supplementary Table 7.

We also checked the mRNA levels of the ethylene-responsive genes in our present study (Supplementary Table 7), and these genes exhibited clear inductions. Besides, the different treatment time of ethylene (4 h in the present study vs 8 h in our previous study) also affects the ethylene-induced intensity of mRNA levels.

b. Second, in ribo-seq experiments it is common that ribosome footprint counts correlate closely with mRNA levels. In this experiment the correlation is extremely low. It appears that nearly every mRNA that is repressed at the transcript level is repressed less (or even induced) at the footprint level, indicating translational enhancement. Together with the previous point, this again suggests that the transcript-level response to ethylene was undercounted.

【Response】 Thanks for your comments. We have reanalyzed our RNA-seq data using a different software DEseq2 with a threshold of genes (both RNA-seq and Ribo-seq reads > 0 RPM vs > 10 RPM in our previous analysis). The results have been integrated into the revised Fig. 6a, b and Supplementary Table 7, 8.

c. Third, the numerical values listed in Line 314 have no relation to the corresponding figure 6a. This is difficult to understand.

【Response】 Thanks for your advice. We have reanalyzed the data and revised the Fig. 6a, b. The corresponding text was also revised in Line 355-363.

d. Fourth, line 317 counts genes with significant changes under ethylene, but the methods are inappropriate. Line 768: The methods say nothing about statistical corrections for multiple testing (false discovery rate analysis). In fact, checking for statistical significance by T-test (line 788) is not appropriate and should not be done. Any claims for statistical significance by t-test are invalid.

【Response】 Thank you for your comments. We reanalyzed the data with the deltaTE to identify the differentially expressed genes at the translation efficiency (TE) level in the revised manuscript. And the statistical corrections for multiple testing (false discovery rate analysis) were also performed and shown in Supplementary Table 7-9. The corresponding text and methods (Line 899-906) were also revised accordingly.

e. Fifth, the data in Fig. S7b look as if they come from a single gene, but it is not listed which. The legend suggests that they are aggregated across all genes. However, the erratic pattern of footprints suggest that this is not the case.

【Response】 Thanks for your comments. The distribution patterns of ribosome footprints for all the genes in each sequenced sample were presented in the revised Supplementary Fig. 11b. Totally results from eight samples were shown.

Taken together, although the ribo-seq figures suggest the very straightforward conclusion that MHZ9 is responsible for the global translational response to ethylene, the underlying data raised a lot of questions that remain to be answered in order to have faith in the result.

【Response】 Thanks for your advice. Based your suggestions, we reanalyzed our RNA-

seq and Ribo-seq data. Based on our new analyses, thousands of genes were significantly and differentially expressed at the mRNA, translational, and translation efficiency (TE) levels in WT in response to ethylene (Fig. 6a-c). And 91% of TE-altered genes induced by ethylene requires MHZ9 function (Fig. 6d). Our new analyses support that MHZ9 is responsible for the major translational response to ethylene.

Fig. 6i. The model. What this model does not show is what happens to the EBF1/2 protein in ethylene. The model suggests that all the EBF1/2 protein present in 'air' disappears extremely rapidly when the cell is exposed to ethylene. How this happens is not at all clear. Most proteins have a half life of at least several hours. During this time translation of new EBF1 protein may indeed be stalled by the translational control mechanism, but the preexisting EBF1 protein would presumably still linger around for many hours and continue to degrade EIL1. But this does not seem to be happening because EIL1 accumulates very fast within 2h of ethylene in an MHZ9 dependent manner (Fig. S6c) while EIL1 mRNA barely changes (Fig. S6d). The model therefore predicts that EBF1/2 is an extremely unstable protein. Has this been shown to be the case? Or is this a hypothesis to be addressed in future work?

【Response】 Thanks for your suggestions. We have discussed the possibilities involving the *OsEBFs* fates in the discussion part in Line 454-462. The model was also revised (Fig. 6j) and the related description has been integrated into our revised manuscript (Line 413).

Minor points:

Did your ribo-seq experiment confirm the prediction from Fig. 5a, that EBF1 has very low translational efficiency?

【Response】 Thanks for your comments. We have repeated the experiments and analyzed the polysome profiling data, and ~30% of the total *OsEBFs* mRNA was recovered in the polysomal fractions 10-16. Based on our ribo-seq data, the translation efficiency (TE) levels of *OsEBF1* and *OsEBF2* in WT are ~0.74 and ~0.47, respectively (Supplementary Table 9).

Fig S1: In panel g the II did not get printed properly.

Fig. S3b: In panel b it is peculiar that IAA20 and ERF2 are not induced by ethylene in the wild type, when in Fig. 1b they were. How to explain this?

【Response】 Thank you for your comments. We have performed additional experiment to detect the expression of ethylene-inducible genes with 10 μ L/L of ethylene for 8 h. The genes were all induced by ethylene in both WT and *MHZ9*-OE plants. The added results have been integrated into the revised Supplementary Fig. 5b. The inconsistency of these results may be owing to the different treatment time used in these two experiments. In Fig. 1b, the seedlings were treated with ethylene for 8 h. And in Fig.

S3b of the previous version, the seedlings were treated with ethylene for 4 h.

Line 720: 'microgram' is probably incorrect.

Line 748: '30mL' is probably incorrect.

Line 776: 'datas'; this word does not exist yet. Why? Because 'Data' is already a plural. Of course, after years of grammatically abusing 'data' as a singular noun, it is only logical that at some point in time 'datas' would appear on the scene.

【Response】 Thanks for your careful reading of our manuscript. We have thoroughly checked and corrected the grammatical errors and typos in the revised manuscript at our best.

Discussion

The discussion section introduces many additional results, which I feel could be held back for another paper once the current paper is sound and published.

【Response】 Thanks for your advice. The related figures and description for the additional results in the discussion section have been removed.

Writing:

While the first part of the results was edited well and easily readable, the later part is written in a more convoluted manner. "The easier it is to read the harder it is to write".

Example for how to edit the writing: "In WT, an obvious reduction of TEs of OsEBF1/2 mRNAs were detected, especially at later fractions after ethylene treatment for 2 h compared to those in air."  Better: In the wild type, ethylene reduced the abundance of EBF1/2 mRNAs in fraction 14, which harbors large polysomes.

There are many passages that deserve editing.

【Response】 Thanks very much for your suggestions. We have carefully polished the language in the revised manuscript.

Thanks very much for your valuable suggestions and we appreciate it very much.

For Reviewer #3:

This manuscript reports the interesting finding of a novel translational regulator called MHZ9, which plays a significant role in ethylene hormone signaling in rice. Ethylene signaling is important in many agronomically important traits in plants, including responses to biotic and abiotic stresses. Significantly, the results in this manuscript connect a key protein in ethylene signaling (EIN2) with the control of downstream gene

expression in ethylene signaling. Moreover, the mechanism involving MHZ9 appears to be novel and may have broader implications for how translation of mRNAs could be regulated.

In general, the experiments appear to be technically sound, but I feel the reporting of the experiments and data could be made clearer as described below. Overall, the manuscript provides a nice story starting with a mutant and leading to gene identification and elucidation of protein function with mechanistic insights.

【 Response 】 Thank you for your positive comments and we appreciate your suggestions to our manuscript.

Suggestions and questions:

As stated in the Abstract, “regulation of ~ 90% of the translationally affected genes by ethylene requires MHZ9 function (lines 33-34)” and in the Discussion, “MHZ9 directly regulates the translation efficiencies (TEs) of its binding targets, and regulation of more than 90% of the genes with ethylene-altered TE levels is dependent on MHZ9 (lines 365-368)” I think the authors could make clearer statements that they believe such regulation is direct, i.e., does not occur via MHZ9’s control of OsEIL1/2 transcription factor protein levels (which was what I thought upon initially reading the manuscript).

On the other hand, it would be helpful if the authors could spell out in the Results and Discussion precisely why they have ruled out the possibility of enhanced transcription/translation due to the accumulation of the OsEIL1/2 transcription factors that activate downstream ethylene-response genes.

【 Response 】 Thank you for the comments. MHZ9 directly binds to 626 genes and, of these, MHZ9 significantly regulates the translation efficiencies (TEs) of 105 genes. Among the 1956 genes with altered TE in response to ethylene, more than 90% (1788) of the genes are directly (105) or indirectly (1683) regulated by MHZ9. It is therefore possible that many of the MHZ9-dependent and TE-altered genes may be indirectly modulated by MHZ9 through OsEIL1/2, and/or other MHZ9-binding genes or via other mechanisms. These analyses have been added to the results in Line 376-382 and the discussion part in Line 414-417.

The epistasis results for MHZ9 and OsEBF1/2 shown in Figure 5f are unexpected. This unexplained finding should be discussed in the Discussion. For example, could there be a role for MHZ9 in the nucleus where EIN2-C is also found? Related to this, on Line 292, I suggest changing “largely insensitive” to “partially insensitive”.

【 Response 】 Thanks for your comments. *OsEBF1/2* mRNAs may not be the only targets for MHZ9, since *Osebf1 Osebf2 mhz9* still exhibits partial insensitivity in both roots and coleoptiles (Fig. 5g). From our CLIP-seq and RNA-IP seq analyses, hundreds

of genes may be associated with MHZ9 under ethylene. Investigation of the MHZ9 binding genes with altered RNA stability and/or translation efficiencies during ethylene response should disclose more regulatory roles of MHZ9 in ethylene signaling. The discussion has been integrated into the discussion section in Line 463-470. Other parts were also revised accordingly.

Line 133-134: The following sentence needs a bit of clarification: “All the results indicate that MHZ9 and OsEIN2 are likely mutually dependent in ethylene signaling.” What is the specific meaning of “mutually dependent”, and what is the rationale for this conclusion? For example, I would expect that ethylene insensitivity from a dominant mutant ethylene receptor would show the same complete insensitivity when paired with the *mhz9* mutation. Would this indicate that the ethylene receptor and MHZ9 are “mutually dependent”?

【Response】 Thank you for your comments. Based on the suggestions, we have modified the description about this result in the revised manuscript (Line 156-158).

For the section on “Genetic interaction of MHZ9 with ethylene signaling components in rice”, was there a technical reason why genetic interactions between MHZ9 and OsCTR1 were not examined? Interaction between MHZ9 and OsCTR1 would make good sense given that the corresponding loss-of-function mutants should have opposite phenotypes.

【Response】 Thanks for your comments. The *OsCTR2* is being knocked out using CRISPR/Cas9 in *mhz9* background. We would be able to address the genetic interactions between MHZ9 and OsCTR2 once the double mutants is available. Due to the time limit, we used another method to pursue their relation.

In our previous study (Zhao et al., 2020, Plant Cell, 32:1626-1643), OsCTR2 phosphorylation status has been used as a biochemical indicator of ethylene response. We performed additional experiments and revealed that ethylene reduced the phosphorylation status of OsCTR2 in both WT and *mhz9* mutant, implying that *MHZ9* mutation does not affect the OsCTR2 signal output (Supplementary Fig. 6). This result most likely reflects that the MHZ9 acts downstream of OsCTR2. We have added this into Supplementary Fig. 6, and the corresponding text was also revised accordingly (Line 146-150).

Along the same lines, the authors could add “either upstream or downstream of OsEIL1/2” at the end of the following sentence (lines 138-139), “From all the genetic analyses, MHZ9 may function downstream of OsERS2, and be required for OsEIN2 and OsEIL1/2 function in ethylene signaling”. This would help to clarify that the exact placement of MHZ9 in the pathway is not fully established by the genetic analyses.

【Response】 Thanks for your advice. We have revised the description based on your suggestions (Line 156-158).

The introduction describes the ethylene signaling pathway as known in Arabidopsis. Since this manuscript focuses on rice, it would be relevant to know the similarities and differences that have been identified in rice compared to the ethylene signaling pathway of Arabidopsis.

【Response】 Thanks for your suggestion. We have revised the MS in the introduction part accordingly based on the comments (Line 68-85).

The gene names of all ethylene signaling components are fully spelled out in the Introduction, with the exception of the subject of this manuscript, “mhz9”. It would be helpful for the reader to know what the name “mhz” stands for.

【Response】 Thanks for your advice. We have added the description of mutant name in the revised manuscript (Line 69-71).

Is there an MHZ9 homolog in Arabidopsis? The answer is not explicitly provided but is relevant to the Discussion on lines 381-388.

【Response】 Thank you for your comments. We have added the phylogenetic analysis of MHZ9 homologous proteins in the revised Supplementary Fig. 4. The text was also revised accordingly (Line 125-127).

I understand that the manuscript is likely streamlined to fit within the word limit for the journal, but in some cases, pertinent information appears to be absent. For example, it is unclear how many times experiments were repeated.

【Response】 Thanks for your comments. We have added the description in the corresponding figure legends.

Line 209: I’m not familiar with the “MS2 tethering assay”. If there is space, perhaps a brief explanation would be helpful in the Results or Methods. I did not see a direct description of the MS2 tethering assay under the Methods section.

【Response】 Thanks for your advice. We have added the explanation of the “MS2 tethering assay” in the revised manuscript (Line 231-234).

Line 245, the “optimized dual luciferase assay” should be explained. What is a dual luciferase assay, and in what way has it been optimized?

【Response】 Thank you for the comment. We have added the explanation of the “optimized dual luciferase assay” in the revised manuscript (Line 283-285).

For the dual luciferase assay, it is unclear to just say “in mhz9 and Osein2” (line 251

plus many other sentences). The use of protoplasts for these experiments needs to be specified, e.g., “in mhz9 and Osein2 protoplasts”. Related to this, throughout the manuscript, the authors refer to just “mhz9”, when it would greatly help the reader to see a noun after “mhz9”, such as “mhz9 protoplasts”, “mhz9 gene”, “mhz9 mutant”, or “mhz9 protein” or “mhz9 plants”. For example, on line 98, please change “The mhz9 was isolated “ to “The mhz9 mutant was isolated.”

【Response】 Thanks for your advice on our manuscript. We have carefully checked the related description and revised them in the revised manuscript.

In Figure 5e, the input amount of protein is not known.

【Response】 Thanks for your comment. We have added the description in the revised legend of Fig. 5f.

It looks like complete statistical analyses are missing for the data in Figure 2b.

【Response】 Thank you for the comment. We have added the statistical analyses in the revised Fig. 2b.

The authors should cite a reference for this portion of line 145: “whose Arabidopsis homologue EIN5 is a P-body marker”, as they did for OsDCP2 on line 149.

【Response】 Thanks for your advice. We have added the related reference in the revised manuscript (Line 170).

In Figure 3f, what do the “x1”, “x10”, and “x10²” refer to? If they indicate dilutions of yeast cells, then the exponent should be a negative number, correct? In the legend, I suggest spelling out “Y2H”.

【Response】 Thank you for the comment. We have made corrections in the revised Fig. 3e. The legend of Fig. 3e was also revised as your suggestions.

In Figure 3i, white asterisks would be easier to see than the red asterisks.

【Response】 Thanks for your advice. We have made modifications in the revised Fig. 3h.

Line 189: Spell out “long intergenic non-coding RNAs” when first using the term “lincRNAs”.

【Response】 Thanks for your comment. We have made modifications in the revised manuscript (Line 211-212).

On line 287, it would be helpful to specify the type of mutants used in the epistasis analysis, such as “single loss-of-function mutants”

【Response】 Thanks for your advice on our manuscript. We have made revisions in the manuscript (Line 334-343).

The grammar in the manuscript needs some improvement, especially with respect to incorrectly leaving out the word “the” in many places or leaving out “a” or “an”.

【Response】 Thanks for your suggestion. We have made corrections in the revised manuscript.

Lines 118 and 122, change to: “field conditions”.

【Response】 Thanks for the comment. We have made corrections in the revised manuscript (Line 137 and 141).

Typos:

Line 420: “MHZ9-regualted”

Line 93, In ,“Our study identifies a master translational regulator of MHZ9”, I believe the word “of” should be deleted, given that MHZ9 itself is the master translational regulator.

【Response】 Thanks for your comment. We have made corrections in the revised manuscript (Line 103 and 418).

Thanks for your careful reading and all your comments on the MS.

REVIEWER COMMENTS

Reviewer #1 (Remarks to the Author):

It is a very interesting and important research to illustrate the function of EIN2 in translational regulation. The authors have addressed all my concerns, I have no further comments.

Reviewer #2 (Remarks to the Author):

The authors have revised their analysis of the role of the new rice MHZ9 gene in the ethylene signaling pathway and specifically its role in translational regulation.

As I pointed out last time, this is a very detailed and thorough paper. The phenotypic characterization of the *mhz9* mutant and its position in the ethylene signaling pathway is very clear. I appreciated the additional information about the numerous *mhz* mutants and ethylene pathway genes in rice.

In keeping with my comments from the first review I will mainly focus on the translation assays. The authors performed several experiments that I had suggested. And these experiments indeed yielded useful results.

Major comment:

1. One notion that needs to be clarified is that MHZ9 has roles that are independent of ethylene.
2. It might also be useful to include some information in the abstract about the target mRNAs of MHZ9, and that the evidence for their translational control by MHZ9 is mixed, that is, dependent on the assay being used. For instance, EBF1/2 is a clear target for translational control by MHZ9 in the reporter gene assay, a weak target in the polysome loading assay and not mentioned as a target in the ribo-seq assay. Also, finding MHZ9 in P-bodies is an important result.

Other comments in order of appearance.

Line 265. MHZ9 has an effect on EBF1 mRNA levels and mRNA stability (Fig. S10a). The effect is opposite from what one might expect from an RNA binding protein that sends certain mRNAs into P-bodies; i.e. MHZ9 slightly stabilized and raised mRNA levels. Given that MHZ9 has hundreds of mRNA targets we should not be too surprised that its effect on EBF1/2 is perhaps indirect.

Line 265. The authors also reanalyzed or repeated the analysis of how MHZ9 affects polyribosome loading of various mRNAs \pm ethylene. First of all, 4h or 8h of ethylene has no effect on global polysome loading in two day old etiolated seedlings (Fig. 5a). This is consistent with the ribo-seq effects (Fig. 6b, c), which include both positive and negative effects. Apparently, the translational repression by EIN2 and MHZ9 is not widespread enough to trigger a global loss of polysome loading, as happens during hypoxia, darkness, heat, etc. However, this result simplifies the following analysis of individual mRNAs. Second, and importantly, for MHZ9's binding targets EBF1 and EBF2, this revised version now shows that ethylene slightly represses polysome loading in a manner that may be MHZ9-dependent (Fig. 5b). This effect is fairly subtle, but this is not out of the ordinary, and it matches the notion that MHZ9 is a translational repressor of EBF1/2.

Line 281. It is good to see these polysome loading experiments reanalyzed in a transparent manner. However, I would like the authors to also grapple with the following result in Fig. 5b: Compared to the weak effect on EBF1/2, MHZ9 has remarkable effects on polysome loading of other mRNAs, specifically eEF1 α , polyubiquitin, and GAPDH! All three are pulled out of the untranslated fraction and into the polysome pool by MHZ9. The authors ignored this important observation. This activity of MHZ9 is not ethylene dependent. However, one can expect that it must be highly regulated, perhaps by light and darkness. After all, if eEF1 α were barely translated at all in *mhz9* plants, as shown here, translation and growth would be severely curtailed, which is not the case in *mhz9* mutants. The authors should discuss this result (also in light of the complementary ribo-seq data).

Line 375. The description and interpretation of the rib-seq data is also a lot clearer now. It appears that many translational effects of ethylene are indeed dependent on MHZ9.

Line 394. The authors should also acknowledge that many of the MHZ9-bound target mRNAs are not regulated translationally by ethylene, and MHZ9 has no effect on their translation (Fig. 6g, h). However, I agree that some of the MHZ9-bound mRNAs are translationally regulated by MHZ9. In these analyses, the statistical power to

detect effects decreases as more pairwise comparisons are piled on top of each other. Footprint counts / mRNA counts  ethylene effects  MHZ9 effect.

Line 395. alternations should be changed to alterations.

Line 443. I take it that the Ribo-Seq experiment did not provide evidence for translational control of EBF1/2 mRNAs by ethylene or MHZ9?

Reviewer #3 (Remarks to the Author):

Thank you for the revisions that have improved the manuscript.

I have only a few small comments on the writing:

Lines 68-85: The new introductory paragraph on what is known in rice is helpful, but it would be even better if the authors synthesized the information for the reader. For example, how do the ethylene signaling pathways in Arabidopsis and rice compare with each other? Are the signaling pathways mostly conserved or quite diverged? Are there novel components/mechanisms in rice ethylene signaling that do not seem to be present in Arabidopsis?

i

Line 76-77: In the new paragraph it isn't clear that MHL1 and MHL2 are Arabidopsis proteins. Perhaps revise the sentence to say something such as, " Similar proteins in Arabidopsis, called MHZ3-LIKE1 (MHL1) and MHL2, have been found to function as positive regulators of ethylene responses in Arabidopsis19."

Line 77, change "Besides" to "In addition"

Line 232: this phrase "the bacteriophage MS2 coat protein (MCP) fused fluorescent protein binds to" should say "...fused to a fluorescent protein....."

Lines 238-239: Either remove the word "the" before "MHZ9", or add the word "protein" after "MHZ9" in the following sentence: "...indicating that the MHZ9 binds to the 3'UTR of OsEBF2 mRNA". The same issue exists elsewhere in the manuscript. That is, if you write "the MHZ9" there should be a noun after "MHZ9", such as "MHZ9 protein". Alternatively, remove the word "the"

Response to the reviewers' comments

For Reviewer #1:

It is a very interesting and important research to illustrate the function of EIN2 in translational regulation. The authors have addressed all my concerns, I have no further comments.

【Response】 Thank you for the comments.

For Reviewer #2:

The authors have revised their analysis of the role of the new rice MHZ9 gene in the ethylene signaling pathway and specifically its role in translational regulation.

As I pointed out last time, this is a very detailed and thorough paper. The phenotypic characterization of the mhz9 mutant and its position in the ethylene signaling pathway is very clear. I appreciated the additional information about the numerous mhz mutants and ethylene pathway genes in rice.

In keeping with my comments from the first review I will mainly focus on the translation assays. The authors performed several experiments that I had suggested. And these experiments indeed yielded useful results.

【Response】 Thank you for the comments.

Major comment:

1. One notion that needs to be clarified is that MHZ9 has roles that are independent of ethylene.

【Response】 Thank you for the comments. We have discussed the possible ethylene-independent roles of MHZ9 in addition to its roles in ethylene signaling in the discussion part in Line 531-547.

2. It might also be useful to include some information in the abstract about the target mRNAs of MHZ9, and that the evidence for their translational control by MHZ9 is mixed, that is, dependent on the assay being used. For instance, EBF1/2 is a clear target for translational control by MHZ9 in the reporter gene assay, a weak target in the polysome loading assay and not mentioned as a target in the ribo-seq assay. Also, finding MHZ9 in P-bodies is an important result.

【Response】 Thank you for the comments. We have added the description of other

MHZ9 binding mRNAs in the abstract. Ribo-seq analysis indicated that, upon 4 h of ethylene treatment, the TE of *OsEBF1* mRNA was significantly inhibited in a MHZ9-dependent manner (Supplementary Table 9). The evidence for the translational control of *OsEBF1/2* by MHZ9 has been added in the discussion part in Line 461-469.

Other comments in order of appearance.

Line 265. MHZ9 has an effect on *EBF1* mRNA levels and mRNA stability (Fig. S10a). The effect is opposite from what one might expect from an RNA binding protein that sends certain mRNAs into P-bodies; i.e. MHZ9 slightly stabilized and raised mRNA levels. Given that MHZ9 has hundreds of mRNA targets we should not be too surprised that its effect on *EBF1/2* is perhaps indirect.

【Response】 Thank you for the comments. Based on the suggestion, we have added the discussion about the effect of MHZ9 on *OsEBFs* mRNA stability in the discussion part in Line 457-461.

Line 265. The authors also reanalyzed or repeated the analysis of how MHZ9 affects polyribosome loading of various mRNAs \pm ethylene. First of all, 4h or 8h of ethylene has no effect on global polysome loading in two day old etiolated seedlings (Fig. 5a). This is consistent with the ribo-seq effects (Fig. 6b, c), which include both positive and negative effects. Apparently, the translational repression by EIN2 and MHZ9 is not widespread enough to trigger a global loss of polysome loading, as happens during hypoxia, darkness, heat, etc. However, this result simplifies the following analysis of individual mRNAs. Second, and importantly, for MHZ9's binding targets *EBF1* and *EBF2*, this revised version now shows that ethylene slightly represses polysome loading in a manner that may be MHZ9-dependent (Fig. 5b). This effect is fairly subtle, but this is not out of the ordinary, and it matches the notion that MHZ9 is a translational repressor of *EBF1/2*.

【Response】 Thank you very much for the comments.

Line 281. It is good to see these polysome loading experiments reanalyzed in a transparent manner. However, I would like the authors to also grapple with the following result in Fig. 5b: Compared to the weak effect on *EBF1/2*, MHZ9 has remarkable effects on polysome loading of other mRNAs, specifically *eEF1alpha*, polyubiquitin, and *GAPDH*! All three are pulled out of the untranslated fraction and into the polysome pool by MHZ9. The authors ignored this important observation. This activity of MHZ9 is not ethylene dependent. However, one can expect that it must be highly regulated, perhaps by light and darkness. After all, if *eEF1alpha* were barely translated at all in *mhz9* plants, as shown here, translation and growth would be severely curtailed, which is not the case in *mhz9* mutants. The authors should discuss this result (also in light of the complementary ribo-seq data).

【Response】 Thank you for the comments. The *OseEF1α* and *OsUBQ5* mRNAs were identified as MHZ9-direct binding mRNA targets (Supplementary Table 10). In our Ribo-seq analysis, the TEs of *OseEF1α* and *OsUBQ5* mRNAs were also slightly reduced in the *mhz9* mutant compared with those in WT in an ethylene-independent manner (Supplementary Table 10). The reason that the poor translation of the *OseEF1α* in *mhz9* mutant does not affect the mutant phenotype was also discussed (Line 535-545). We have made the revisions in text (Line 296-297, 299) and discussed the ethylene-independent effect of MHZ9 on the translation of these mRNAs in the discussion part in Line 531-547.

Line 375. The description and interpretation of the rib-seq data is also a lot clearer now. It appears that many translational effects of ethylene are indeed dependent on MHZ9.

【Response】 Thank you for the comments.

Line 394. The authors should also acknowledge that many of the MHZ9-bound target mRNAs are not regulated translationally by ethylene, and MHZ9 has no effect on their translation (Fig. 6g, h). However, I agree that some of the MHZ9-bound mRNAs are translationally regulated by MHZ9. In these analyses, the statistical power to detect effects decreases as more pairwise comparisons are piled on top of each other. Footprint counts / mRNA counts  ethylene effects  MHZ9 effect.

【Response】 Thank you for the comments. Based on the suggestions, we have made revisions in the text (Line 409-411) about this result. The discussion part was also revised accordingly at Lines 531-547.

Line 395. alternations should be changed to alterations.

【Response】 Thanks for the comment. We have made corrections in the revised manuscript (Line 412).

Line 443. I take it that the Ribo-Seq experiment did not provide evidence for translational control of *EBF1/2* mRNAs by ethylene or MHZ9?

【Response】 Thanks for the comment. In the ribo-seq assay, upon 4 h of ethylene treatment, the TE of *OsEBF1* mRNA was significantly inhibited in a MHZ9-dependent manner (Supplementary Table 9). We have made revisions in the text (Line 386) and discussion part (Line 467-469).

For Reviewer #3:

Thank you for the revisions that have improved the manuscript.

【Response】 Thanks for the comment.

I have only a few small comments on the writing:

Lines 68-85: The new introductory paragraph on what is known in rice is helpful, but it would be even better if the authors synthesized the information for the reader. For example, how do the ethylene signaling pathways in Arabidopsis and rice compare with each other? Are the signaling pathways mostly conserved or quite diverged? Are there novel components/mechanisms in rice ethylene signaling that do not seem to be present in Arabidopsis?

【Response】 Thanks for your advice. We have revised the description based on your suggestions (Line 72-95).

Line 76-77: In the new paragraph it isn't clear that MHL1 and MHL2 are Arabidopsis proteins. Perhaps revise the sentence to say something such as, " Similar proteins in Arabidopsis, called MHZ3-LIKE1 (MHL1) and MHL2, have been found to function as positive regulators of ethylene responses in Arabidopsis19."

【Response】 Thanks for your comments. We have revised the description based on your suggestions (Line 96-97).

Line 77, change "Besides" to "In addition"

【Response】 Thanks for the comment. The corresponding part has been revised accordingly.

Line 232: this phrase "the bacteriophage MS2 coat protein (MCP) fused fluorescent protein binds to" should say "...fused to a fluorescent protein....."

【Response】 Thanks for the comment. We have made corrections in the revised manuscript (Line 248).

Lines 238-239: Either remove the word "the" before "MHZ9", or add the word "protein" after "MHZ9" in the following sentence: "...indicating that the MHZ9 binds to the 3' UTR of OsEBF2 mRNA". The same issue exists elsewhere in the manuscript. That is, if you write "the MHZ9" there should be a noun after "MHZ9", such as "MHZ9 protein". Alternatively, remove the word "the"

【Response】 Thanks for your advice on our manuscript. We have carefully checked the related description and revised them in the manuscript.

REVIEWERS' COMMENTS

Reviewer #2 (Remarks to the Author):

I do not have any further comments. I enjoyed reviewing this piece of work.